# Latent Wasserstein Adversarial Imitation Learning

## Abstract

Imitation Learning (IL) enables agents to mimic expert behavior by learning from demonstrations. However, traditional IL methods require large amounts of medium-to-high-quality demonstrations as well as actions of expert demonstrations, both of which are often unavailable. To address these limitations, we propose LWAIL (Latent Wasserstein Adversarial Imitation Learning), a novel adversarial imitation learning framework that focuses on state-only distribution matching by leveraging the Wasserstein distance computed in a latent space. To obtain a meaningful latent space, our approach includes a pre-training stage, where we employ the Intention Conditioned Value Function (ICVF) model to capture the underlying structure of the state space using randomly generated state-only data. This enhances the policy's understanding of state transitions, enabling the learning process to use only one or a few state-only expert episodes to achieve expert-level performance. Through experiments on multiple MuJoCo environments, we demonstrate that our method outperforms prior Wasserstein-based IL methods and prior adversarial IL methods, achieving better sample efficiency and policy robustness across various tasks.

## 1 Introduction

As a powerful tool for solving sequential decision-making problems, Reinforcement Learning (RL) has achieved remarkable success in recent years across various fields, such as gaming (Silver et al., 2016) and training of large language models from human feedback (Ramamurthy et al., 2023). However, RL relies heavily on well-defined reward signals (Li et al., 2021), which can be difficult to obtain in real-world settings (e.g., robot control (Ibarz et al., 2021) with varied target tasks) or may require careful, environment-specific considerations (Yu et al., 2020).

Imitation Learning (IL) provides a way to avoid the use of rewards, and can generally be divided into two types: Behavioral Cloning (BC) (Ross et al., 2011; Torabi et al., 2018a) and Inverse Reinforcement Learning (IRL) (Ho & Ermon, 2016; Fu et al., 2018; Bobrin et al., 2024). Compared to BC, which directly learns to map states to actions by imitating an expert, inverse reinforcement learning is more flexible and robust as it recovers a (usually dense) reward function from existing demonstrations (Arora & Doshi, 2021) that exhibit the essence of the target policy.

However, similar to well-defined reward signals, expert demonstrations are often sparse in real-life applications too, as human effort is usually involved (e.g., teleoperating robotic arms) whenever a new task is considered. In this paper, we focus on one type of expert data shortage that has been widely studied by the community: *state-only* expert demonstrations, which is also referred to as 'Imitation Learning from Observations' (LfO). Classic IRL-based methods, such as GAIL (Ho & Ermon, 2016) and AIRL (Fu et al., 2018), require access to expert actions, which are unavailable in LfO. Existing methods for LfO (Torabi et al., 2018b; Zhu et al., 2020), similar to GAIL, train a discriminator to distinguish an expert's behavior observed via expert demonstrations from the agent's behavior encoded in the learned policy.

There are also recent LfO methods minimizing state (Ma et al., 2022) or adjacent state-pair (Liu et al., 2020; Kim et al., 2022a) occupancy divergence. They often require large amounts of medium-to-high-quality offline data to perform well. In contrast, random data are much easier to obtain from suboptimal agents, exploratory behaviors, or even failed attempts at completing a task. Thus, we

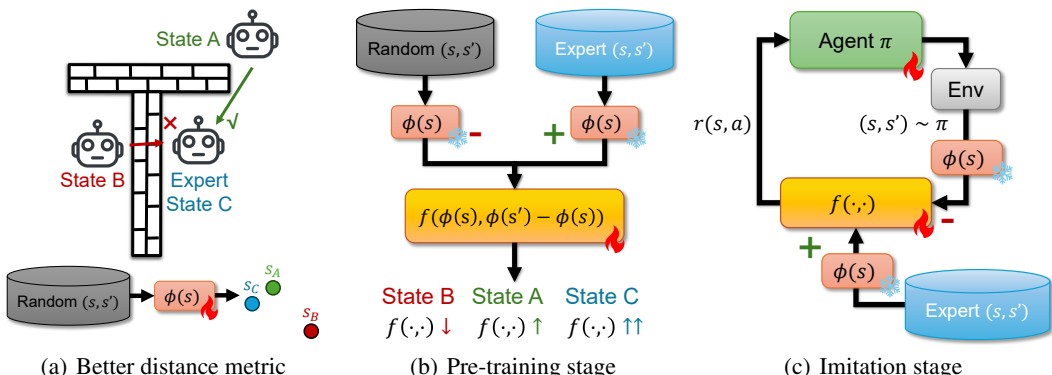

(a) Better distance metric     (b) Pre-training stage     (c) Imitation stage

Figure 1: Illustrating our motivation for a better distance metric and an outline of the algorithm. Panel a) illustrates a case where the Euclidean distance between states is not a good metric: state B is closer to expert state C, but it is apparently less desirable than more distant state A, as it cannot reach C. To address this, we use random data and ICVF to find a more meaningful embedding space as shown in the lower half. Panel b), together with the lower half of panel a), shows our pre-training stage: we first train ICVF to obtain $\phi(s)$, which serves as a reward for our agent in the online stage shown in panel c). Flames indicate training weights and snowflakes indicate frozen weights.

wonder: *can we learn a good policy from a few (preferably a single) state-only expert demonstrations, optionally very low (random) quality offline data, and online interaction?*

To answer this question, we propose an adversarial imitation learning approach for state-only distribution matching using the Wasserstein distance. Different from prior work, we improve upon the classic and default Euclidean distance metric in the Wasserstein formulation by incorporating an embedding model which is trained prior to policy optimization with a large amount of random data via the Intention Conditioned Value Function (ICVF) (Ghosh et al., 2023). Using it, we acquire a distance metric which captures the dynamics and hidden relationships between states. Compared to prior adversarial imitation learning methods such as GAIL (Ho & Ermon, 2016) and AIRL (Fu et al., 2018) which often use state-action pairs to learn, our proposed method needs much less data; compared to prior Wasserstein-based imitation learning methods such as WDAIL (Zhang et al., 2020b) and IQlearn (Garg et al., 2021), our learned distance metric improves upon the classic Euclidean distance inherently applied by the popular Kantorovich-Rubinstein (KR) duality (see Sec. 3.2 for details) without using surrogates such as the one in PWIL (Dadashi et al., 2021).

To validate our approach, we conduct experiments in two settings. First, we validate our method on Maze2D environments from the D4RL benchmark (Fu et al., 2020). This setting provides the added advantage of compelling visualizations, enabling a clear validation of our idea. Further, we extend our experiments to challenging locomotion tasks in the MuJoCo environment. We train our ICVF distance metric model via random data, both from the existing D4RL dataset and data we collected through random actions. We find that the ICVF-learned metric grasps the reachability relations within a trajectory much better than the Euclidean distance. In terms of imitation learning we achieve strong results using only a single trajectory of state-based expert data.

Our contributions can be summarized as follows: 1) We conceptually and empirically show that the latent space from ICVF provides a good metric for state-based Wasserstein occupancy matching, circumventing an inherent problem of the KR duality; 2) We propose a simple but effective method that can achieve expert level performance with a single state-only expert trajectory; 3) We empirically show that our method outperforms a variety of baselines on multiple testbeds, thus proving a better distance metric can greatly benefit Wasserstein-based adversarial imitation learning.

## 2 PRELIMINARIES

**Markov Decision Process (MDP).** A Markov Decision Process (MDP) is a mathematical framework which describes the interactions of an agent with an environment at discrete time steps. It is defined by the tuple $(\mathcal{S}, \mathcal{A}, P, R, \gamma)$, where $\mathcal{S}$ represents the state space, and $\mathcal{A}$ denotes the action

space. The state transition probability function $P(s'|s,a)$ defines the likelihood of transitioning to a new state $s' \in \mathcal{S}$ after taking an action $a \in \mathcal{A}$ in the current state $s \in \mathcal{S}$. The reward function $R(s,a) \in \mathbb{R}$ specifies the immediate reward received after taking action $a$ in state $s$. $\gamma \in [0,1)$ is the discount factor, determining the importance of future rewards relative to immediate ones.

At each time step $t$, the agent observes the current state $s_t \in \mathcal{S}$, selects an action $a_t \in \mathcal{A}$, receives a reward $r_t = R(s_t, a_t)$, and transitions to the next state $s_{t+1}$ according to the transition function $P$. A complete running process is called a *trajectory*. The goal of the agent is to learn a policy $\pi : \mathcal{S} \to \mathcal{A}$ that maximizes the expected cumulative discounted reward defined as: $G_t = \sum_{k=0}^{\infty} \gamma^k r_{t+k}$. In this paper, we focus on the *state and state-pair occupancy*, which are the visitation frequency of states and state-pairs. Given policy $\pi$, the state occupancy is defined as $d_s^\pi(s) = (1-\gamma) \sum_{t=0}^{\infty} \gamma^t \mathrm{Pr}(s_t = s)$ and the state-pair occupancy is given by $d_{ss}^\pi(s,s') = (1-\gamma) \sum_{t=0}^{\infty} \gamma^t \mathrm{Pr}(s_t = s, s_{t+1} = s')$.

**Wasserstein Distance.** Wasserstein distance, also known as Earth Mover's Distance (EMD) (Kantorovich, 1939), is widely used to measure the distance between two probability distributions. For the metric space $(M, c)$ where $M$ is a set and $c : M \times M \to \mathbb{R}$ is a metric, the 1-Wasserstein distance[1] between two distributions $p(x)$ and $q(x)$ on the metric space $(M, c)$ is defined as:

$$\mathcal{W}_1(p, q) = \inf_{\Pi(p,q)} \int_{M \times M} c(x, y) \, d\Pi(x, y). \tag{1}$$

Intuitively, this equation quantifies the optimal way to "move" mass from $p$ to $q$ while minimizing the total movement, as described by the joint distributions $\Pi(p, q)$ with marginals $p$ and $q$. A more popular form adopted by the machine learning community is the Kantorovich-Rubinstein (KR) dual (Kantorovich & Rubinstein, 1958) of the 1-Wasserstein distance which reads as follows:

$$\mathcal{W}_1(p, q) = \sup_{\|f\|_L \leq 1} \left( \mathbb{E}_{x \sim p}[f(x)] - \mathbb{E}_{x \sim q}[f(x)] \right). \tag{2}$$

Here, $\|f\|_L \leq 1$ restricts function $f$ to be 1-Lipschitz, i.e., for any $x, x'$, $\frac{|f(x) - f(x')|}{c(x, x')} \leq 1$. As the most prominent way to compel Lipschitzness is regularization of the gradient (Gulrajani et al., 2017; Stanczuk et al., 2021) (i.e., $\nabla f(x_0) = \frac{f(x) - f(x_0)}{\|x - x_0\|_2}$ for local $x$), the 1-Lipschitz constraint inherently limits the distance metric $c$ to be Euclidean (Stanczuk et al., 2021), which is often undesirable (Yan et al., 2024). In this paper, we fix this issue by introducing an ICVF-learned distance metric.

**TD3.** Twin Delayed Deep Deterministic Policy Gradient (TD3) (Fujimoto et al., 2018) extends the Deep Deterministic Policy Gradient (DDPG) (Lillicrap et al., 2016) algorithm, designed to mitigate the overestimation bias commonly found in Q-learning. TD3 introduces three key modifications:

1) *Clipped Double Q-learning*: TD3 maintains two Q-networks, $Q_{\theta_1}$ and $Q_{\theta_2}$ parameterized by $\theta_1$ and $\theta_2$ respectively, and uses the smaller of the two as the critic loss to reduce overestimation (Lee & Lee, 2023). More specifically, the fitting target $y$ is calculated by

$$y = r + \gamma \min(Q_{\theta_1}(s', \pi_\phi(s')), Q_{\theta_2}(s', \pi_\phi(s'))),$$

where $\pi_\phi$ is the policy $\pi$ parameterized by $\phi$. The clipped value stabilizes training and results.

2) *Delayed Policy Updates*: To further stabilize learning, the policy is updated less frequently than the critic, reducing the chance of policy updates based on inaccurate Q-values.

3) *Target Policy Smoothing*: To address overfitting to deterministic policies, when calculating the target for the critic loss, a Gaussian noise $\epsilon$ with variance $\sigma^2 > 0$ is clipped with a threshold $c_0 > 0$ before being added to the target action $a'$. More specifically, we have

$$a' = \pi_\phi(s') + \epsilon, \quad \epsilon \sim \mathrm{clip}(\mathcal{N}(0, \sigma^2), -c_0, c_0).$$

This regularizes the policy, making it more robust to small state changes.

TD3 improves upon DDPG and works well particularly in high-dimensional continuous action spaces. In this work, we adopt TD3 for the downstream RL component of our method.

---

[1]Unless otherwise specified, we will discuss 1-Wasserstein distance in this paper.

## 3 LATENT WASSERSTEIN ADVERSARIAL IMITATION LEARNING

This section is organized as follows: In Sec. 3.1 we first define our goal and frame it using a Wasserstein adversarial state occupancy matching objective with KR duality. We then point out its inherent shortcomings and propose the ICVF-trained latent space metric as a solution in Sec. 3.2. Finally, we introduce our algorithm in Sec. 3.3. See Fig. 1 for an overview of our work.

### 3.1 WASSERSTEIN ADVERSARIAL STATE OCCUPANCY MATCHING

Our goal is to learn a good policy $\pi$ using three sources of information: a few-shot, state-only expert dataset $E$, a dataset $I$ with state-only *random* transitions $(s, s')$ (either given or collected with an untrained policy), and online interactions. Inspired by recent state occupancy matching works (Kostrikov et al., 2020; Garg et al., 2021; Ma et al., 2022; Kim et al., 2022a), in this paper, we aim to minimize the 1-Wasserstein distance between state-pair occupancy distributions of the policy $\pi$, i.e., $d_{ss}^{\pi}(s, s')$, and of the empirical policy of the expert, i.e., $d_{ss}^{E}(s, s')$. Formally, we address

$$\min_{\pi} \mathcal{W}_1(d_{ss}^{\pi}(s, s'), d_{ss}^{E}(s, s')), \tag{3}$$

where $s$ and $s'$ are adjacent states in a trajectory. Note, while many occupancy matching works such as SMODICE (Ma et al., 2022) and LobsDICE (Kim et al., 2022a) use $f$-divergences, we opt to use the 1-Wasserstein distance because it provides a smoother measure and leverages the underlying geometric property of the state space, unlike $f$-divergences.

However, the Wasserstein distance itself is hard to compute as it is inherently a constrained linear programming problem (see Eq. (1)), which is difficult to solve via gradient descent. While there exist workarounds such as convex regularizers (Yan et al., 2024), surrogates (Dadashi et al., 2021), and direct matching on trajectories (Luo et al., 2023; Bobrin et al., 2024), here, we choose the widely adopted KR dual (Kantorovich & Rubinstein, 1958) as our objective. Combined with policy optimization, our final objective reads as follows:

$$\min_{\pi} \max_{\|f\|_L \leq 1} \left( \mathbb{E}_{(s,s') \sim d_{ss}^{\pi}}[f(s, s')] - \mathbb{E}_{(s,s') \sim d_{ss}^{E}}[f(s, s')] \right), \tag{4}$$

where the 1-Lipschitz constraint can be encouraged by prominent methods such as gradient regularization (Gulrajani et al., 2017). With constraint 'addressed,' Eq. (4) is a bi-level optimization and can be optimized iteratively by any RL algorithm. Specifically, since $d_{ss}^{E}$ is independent of $\pi$, the objective for finding policy $\pi$ is $\max_{\pi} \mathbb{E}_{(s,s') \sim d_{ss}^{\pi}}[-f(s, s')]$. This can be optimized with any RL algorithm using reward $r(s, a) = \mathbb{E}_{s' \sim P(s'|s,a)}[-f(s, s')]$. From an adversarial imitation learning perspective, $f(s, s')$ can be interpreted as a discriminator that outputs a high score $f(s, s')$ for expert state pairs and a low score $f(s, s')$ for non-expert ones.

### 3.2 OVERCOMING METRIC LIMITATIONS

While the objective mentioned in Eq. (4) already provides a viable solution, it has a subtle limitation: As mentioned in Sec. 2, the metric $c(s, s')$ is limited to be Euclidean in practice due to the Lipschitz constraint $\frac{|f(x) - f(x')|}{c(x, x')} \leq 1$. However, as illustrated in Fig. 1, for complex locomotion tasks, a Euclidean distance in the raw state space often fails to capture the true relation between states due to the high-dimensional and intricate nature of the state representation. This subtle reliance on the Euclidean distance is overlooked by prior KR duality-based imitation learning methods such as IQlearn (Garg et al., 2021) and WDAIL (Zhang et al., 2020b).

To address this issue, we aim to learn a latent state representation, where the Euclidean distance within the latent space serves as a more effective metric. Intuitively, the latent space permits to capture the environment's dynamics and relationship between states from randomly collected, unlabeled data, even without access to ground-truth actions and rewards. To achieve this goal without relying on high quality offline data, we benefit from the Intention Conditioned Value Function (ICVF) (Ghosh et al., 2023) framework, which has provided a compelling representation of the value function. ICVF introduced the concepts of *intention* to replace the traditional actions, and can be defined as the un-normalized likelihood of achieving an outcome state $s_+$ in the future when the

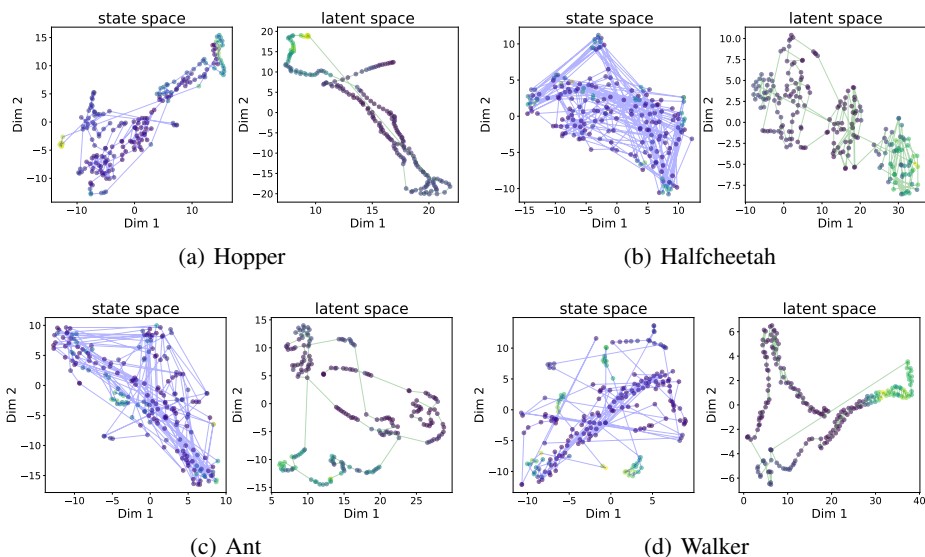

Figure 2: Visualization of the same trajectory in the original state space and the embedding (latent) space. The color of the points represents the ground truth reward of the state. We observe that an ICVF-trained embedding provides a much more dynamics-aware metric than the vanilla Euclidean distance.

agent acts according to intention $z$ starting from a state $s$. More specifically, the value function can be formulated as:

$$V(s, s_+, z) = \mathbb{E}_{s_{t+1} \sim P_z(\cdot|s_t)} \left[ \sum_t \gamma^t \mathbb{I}(s_t = s_+) \mid s_0 = s \right], \tag{5}$$

where $\mathbb{I}(\text{condition})$ is 1 if the condition is true, and 0 otherwise; $P_z(s_{t+1}|s)$ is the transition probability from $s_t$ to $s_{t+1}$ when acting according to intent $z$. This objective can be learned from a random state-action dataset $I$ with offline RL. In this paper, we use Implicit Q-Learning (IQL) (Kostrikov et al., 2022) with the following critic objective where $\alpha \in (0.5, 1]$, $V_{\text{target}}$ is the target function and $A$ is the "advantage" of the current value function (see Appendix A for details):

$$\mathcal{L}(V_\theta) = \mathbb{E}_{(s,s'),z,s_+} \left[ |\alpha - \mathbb{I}(A < 0)| \cdot (V_\theta(s, s_+, z) - \mathbb{I}(s = s_+) - \gamma V_{\text{target}}(s', s_+, z))^2 \right]. \tag{6}$$

Note, the ICVF value function is designed to be structured as follows:

$$V(s, s_+, z) = \phi(s)^T T(z) \psi(s_+), \tag{7}$$

where $\phi(s) \in \mathbb{R}^d$ is the *state representation* that maps a state into a latent space, $T(z) \in \mathbb{R}^{d \times d}$ is the matrix of *counterfactual intention*, and $\psi(s_+) \in \mathbb{R}^d$ is the *outcome representation* (see Appendix A for details).

Prior work (Yan et al., 2024) has shown that selecting a good metric is crucial for the performance of Wasserstein-based solutions. Importantly, Euclidean distance $c(s, s') = \|\phi(s) - \phi(s')\|_2$ in the latent space can serve as a more suitable Wasserstein distance metric, capturing the structure of the environment (Bobrin et al., 2024) more faithfully. To better show this, we provide a t-SNE (Van der Maaten & Hinton, 2008) visualization of the same trajectory in both the raw state space and the latent space in Fig. 2. The result shows that the latent space better captures the dynamic relationship between states. This finding highlights that the Euclidean distance in this space is a more suitable metric for Wasserstein distance-based state matching.

## 3.3 LWAIL

In this subsection, we introduce the pipeline of our proposed method, LWAIL, which consists of two stages: pre-training and imitation. We will also provide pseudo-code to summarize our approach.

---

**Algorithm 1** LWAIL

---

**Require:** State-only expert dataset $E$, state-action random dataset $I$ (optional), initial policy $\pi$, discriminator $f$, replay buffer $\mathcal{B}$, update frequency $m$
    **Pretrain:**
1: Collect transitions into buffer $\mathcal{B}$ with random actions or use random dataset provided
2: Use ICVF to pre-train the representation network $\phi$ (Eq. (6))
3: Pre-train $f$ with initial policy $\pi$ (inner level of Eq. (9))
4: **Imitation:**
5: **while** $t \leq T$ **do**
6:     Collect transitions $(s, a, s', \text{done})$ using $\pi$
7:     Calculate pseudo-reward $r_p = \sigma(-f(\phi(s), \phi(s') - \phi(s)))$
8:     Add $(s, a, s', r_p, \text{done})$ to replay buffer $\mathcal{B}$
9:     **if** $t \mod m == 0$ **then**
10:       Update $f$ (inner level of Eq. (9))
11:     **end if**
12:     Sample mini-batch of $N$ transitions from $\mathcal{B}$ to perform TD3 update
13: **end while**

---

**Pre-training.** The pre-training stage contains three steps. The first step is to collect random transition data using a randomly initialized policy within the environment. This step can be skipped if a random dataset is available, e.g., in a setting following Yue et al. (2024). The second step involves training of ICVF following Eq. (6) with IQL. We then retrieve the projection function $\phi$ from Eq. (7). Finally, the third step is to train $f(\cdot, \cdot)$ using the following objective with frozen latent variable mapping $\phi$ and frozen, untrained, random policy $\pi$:

$$\max_{\|f\|_L \leq 1} \mathbb{E}_{(s,s') \sim d_{ss}^\pi}[f(\phi(s), \phi(s') - \phi(s))] - \mathbb{E}_{(s,s') \sim d_{ss}^E}[f(\phi(s), \phi(s') - \phi(s))]. \tag{8}$$

Here, $f$ serves as the discriminator (from an adversarial learning perspective), the reward function for the policy in later imitation (from an RL perspective) and the KR dual function (from a Wasserstein perspective). To encourage the Lipschitz constraint, $f$ is trained with a gradient penalty, following WGAN-GP (Gulrajani et al., 2017). Note, instead of directly operating on $\phi(s')$, we use $\phi(s') - \phi(s)$ as the second input to $f$ (which also applies for offline initialization of $f$), allowing the discriminator to better learn the difference between expert and non-expert transitions.

**Imitation.** In the (online) imitation learning stage, we again freeze the ICVF-learned embedding $\phi$ and replace $s$ and $s'$ with their latent space representations, $\phi(s)$ and $\phi(s')$. Then the imitation learning problem in Eq. (4) can be addressed via

$$\min_\pi \max_{\|f\|_L \leq 1} \mathbb{E}_{(s,s') \sim d_{ss}^\pi}[f(\phi(s), \phi(s') - \phi(s))] - \mathbb{E}_{(s,s') \sim d_{ss}^E}[f(\phi(s), \phi(s') - \phi(s))]. \tag{9}$$

Following the standard off-policy approach, the agent interacts with the environment to gather data and iteratively updates the value function and policy. Once the policy has collected a batch of trajectories, we update the discriminator network $f$ based on Eq. (9). Following Sec. 3.1, using an adversarial learning framework, we then use $f$ to generate rewards for the downstream reinforcement learning algorithm, for which we employ TD3 (Fujimoto et al., 2018), a robust, off-policy reinforcement learning method selected due to its stability and effectiveness. Slightly different from Sec. 3.1 however, the reward for the TD3 policy $\pi$ is defined as $r(s, s') = \sigma(-f(\phi(s), \phi(s') - \phi(s)))$. $\sigma$ is the sigmoid function that normalizes the reward to the range $[0, 1]$, stabilizing the downstream RL algorithm. $-f(\phi(s), \phi(s') - \phi(s))$ is a 1-sample estimation of $\mathbb{E}_{s' \sim P(s'|s,a)}[-f(\phi(s), \phi(s') - \phi(s))]$ for transition $(s, a, s')$, following Kim et al. (2022a). $f$ and $\pi$ are then iteratively updated until the policy $\pi$ is properly trained. The entire procedure of our method is summarized in Alg. 1.

## 4 EXPERIMENTS

In this section, we assess efficacy of LWAIL across multiple benchmark tasks. Specifically, we want to verify the following questions: 1) How is our assigned reward $f(\phi(s), \phi(s') - \phi(s))$ different from the ground-truth reward? 2) Can our algorithm work well on complicated continuous control

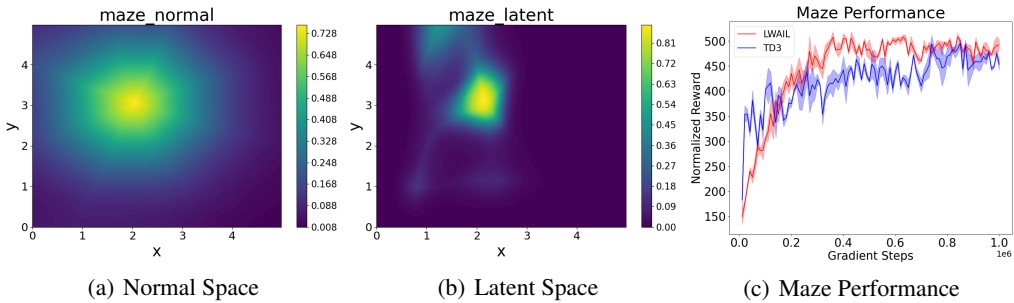

(a) Normal Space        (b) Latent Space        (c) Maze Performance

Figure 3: Results on the Maze2D environment. Panel a) and panel b) illustrate the reward distribution in the Maze2D environment, without and with the ICVF embedding. Panel c) shows the performance of our method and TD3 with ground truth sparse reward in this environment, using the normalized score as the evaluation metric.

environments? 3) How much do the components of LWAIL contribute to its performance? We will answer 1) in Sec. 4.1, 2) in Sec. 4.2, and 3) in Sec. 4.3 respectively.

### 4.1 SIMPLE ENVIRONMENT ON MAZE2D

We first evaluate LWAIL in the Maze2D environment, offering clear visualizations for better understanding of our method; specifically, how LWAIL effectively learns reward representations for downstream tasks.

**Experimental and Dataset Setup.** We use the maze2d-open-v0 environment from D4RL (Fu et al., 2020), a simple setting where a 2D point mass is guided from a random starting point to a specific target at coordinates (2,3). The observation space is 4-dimensional, consisting of the $x$ and $y$ coordinates of the point mass's position and its $x$ and $y$ velocities. Actions correspond to linear forces applied to the point mass in the $x$ and $y$ directions. The sparse reward is 1 if the ball is in the final target position (the Euclidean distance between the ball and the goal is lower than 0.5 m). The ICVF model is trained using the D4RL random dataset from maze2d-open-v0. Next, we apply the standard Wasserstein learning method using both Euclidean distance and a latent space representation.

**Results.** After convergence, the resulting reward map is shown in Fig. 3 a) and b), where the input state used to calculate rewards is the position with zero velocity. These results demonstrate that the ICVF-learned metric provides a more distinctive reward signal that is trajectory-aware, and such awareness of trajectory dynamics improves reward signal feedback quality during the exploration process of online inverse RL. We further plot the reward curve in Fig. 3 c) to show that our method converges to a good result on Maze2d, outperforming TD3 with ground-truth sparse reward.

### 4.2 MUJOCO ENVIRONMENT

**Baselines.** We test a variety of baselines in this section, which can be categorized into four types: 1) *classic imitation methods*, including GAIL (Ho & Ermon, 2016), AIRL (Fu et al., 2018) and the plain Behavior Cloning (BC); 2) *Wasserstein-based imitation methods*, including PWIL (Dadashi et al., 2021), WDAIL (Zhang et al., 2020b) and IQlearn (Garg et al., 2021); 3) *LfO methods*, including BCO (Torabi et al., 2018a), GAIfO (Torabi et al., 2018b), DACfO (LfO variant of (Kostrikov et al., 2019) serves as a baseline in Zhu et al. (2020)) and OPOLO (Zhu et al., 2020); 4) *offline to online imitation learning*, which includes OLLIE (Yue et al., 2024).[2] Some methods such as GAIL and WDAIL require expert action; for these methods, we report the results with extra access to the expert actions. We report mean and standard deviation from 5 independent runs with different seeds. The performance is measured by the normalized reward defined in the D4RL dataset (higher is better).

**Experimental and Dataset Setup.** We test our method on four standard MuJoCo (Todorov et al., 2012) environments: hopper, walker2d, halfcheetah and ant. The expert data for our method and

---

[2]We are unable to run the github version of OLLIE due to non-trivial typos in their code. We directly report the final numbers for random dataset from their paper instead.

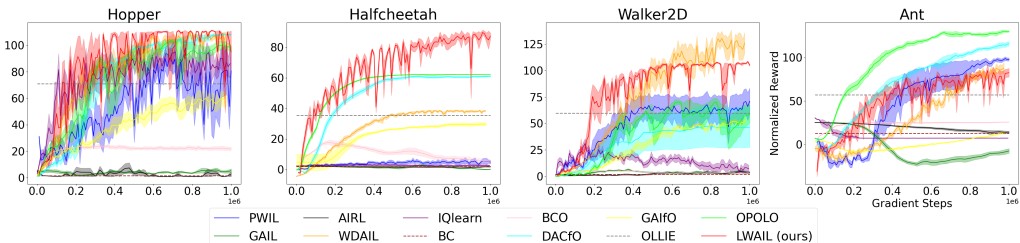

Figure 4: Performance comparison on the MuJoCo environments. Our method generally shows compelling results among all environments and baselines. Gradient steps is equal to online samples in our method.

| | Hopper | HalfCheetah | Walker | Ant | Average |
|---|---|---|---|---|---|
| OLLIE* | $71.10 \pm 3.5$ | $35.50 \pm 4.0$ | $59.80 \pm 8.5$ | $57.10 \pm 7.0$ | 55.87 |
| BC* | $1.51 \pm 0.61$ | $1.92 \pm 0.80$ | $2.06 \pm 0.99$ | $12.74 \pm 2.34$ | 4.56 |
| GAIL* | $7.78 \pm 2.13$ | $-0.33 \pm 0.60$ | $2.14 \pm 1.03$ | $-1.98 \pm 4.41$ | 1.90 |
| AIRL* | $1.16 \pm 0.43$ | $6.02 \pm 3.59$ | $0.83 \pm 0.95$ | $3.30 \pm 11.39$ | 2.83 |
| WDAIL* | $107.72 \pm 4.70$ | $38.30 \pm 1.09$ | $\mathbf{126.07 \pm 19.36}$ | $87.59 \pm 13.12$ | 89.92 |
| PWIL | $78.93 \pm 39.00$ | $20.81 \pm 33.21$ | $84.01 \pm 26.53$ | $105.62 \pm 2.36$ | 72.34 |
| IQlearn | $86.24 \pm 21.92$ | $2.51 \pm 1.05$ | $7.07 \pm 7.23$ | $7.39 \pm 0.13$ | 25.80 |
| BCO | $21.31 \pm 4.06$ | $4.08 \pm 1.72$ | $0.88 \pm 0.85$ | $25.33 \pm 0.87$ | 12.90 |
| DACfO | $109.46 \pm 0.39$ | $61.52 \pm 0.76$ | $45.28 \pm 37.26$ | $113.40 \pm 10.20$ | 82.41 |
| GAIfO | $58.74 \pm 9.07$ | $29.79 \pm 2.12$ | $52.73 \pm 4.16$ | $12.99 \pm 2.77$ | 38.56 |
| OPOLO | $99.24 \pm 5.49$ | $58.98 \pm 7.46$ | $37.07 \pm 12.67$ | $\mathbf{129.46 \pm 3.64}$ | 81.19 |
| LWAIL (ours) | $\mathbf{110.52 \pm 1.06}$ | $\mathbf{86.71 \pm 5.67}$ | $105.30 \pm 2.33$ | $80.56 \pm 13.09$ | $\mathbf{95.77}$ |

Table 1: Performance comparison on the MuJoCo environments. Here, "*" represents methods with extra access to expert actions. Results are averaged over 50 trajectories. It is apparent that our method outperforms most baselines, even those with access to expert actions. DACfO and OPOLO are the most competitive expert state-only baselines.

baselines is one trajectory from the D4RL expert dataset. The random data used for ICVF pre-training is from the D4RL random dataset. We use a single state-only trajectory as the expert data for all the baselines. We use normalized average reward from 10 evaluation trajectories as our metric and report its mean and standard deviation (higher reward means better results). We train our method and baselines for 1M online samples. See Appendix B.3 for more hyperparameters.

**Results.** Fig. 4 shows the reward curve of each methods on MuJoCo environments, while Tab. 1 summarizes the final results. Both figure and table show that our method achieves compelling results and convergence compared to the baselines across tasks. BCO, DACFO, OPOLO and PWIL perform well on some environments, while other methods struggle.

### 4.3 ABLATION STUDY

**Is our learned reward better than the ground truth reward?** We compare our method with ground truth guided TD3. Mean and standard deviation are obtained from 3 independent runs with different seeds. The result is illustrated in Fig. 5. It empirically shows that LWAIL with our learned reward can achieve comparable or better performance than a human designed ground truth reward.

**Is ICVF-learned embedding helping the performance?** We compare the performance between our original method and our method without ICVF-learned embedding, that is, we get a pseudo reward generated by $f(s, s' - s)$ without embedding $\phi$. Mean and standard deviation are obtained from 3 independent runs with different seeds. The result is in Fig. 6. We observe that agents without ICVF-learned embedding tend to remain in stable but relatively low-reward states, exhibiting less tendency to improve due to less positive feedback in exploration.

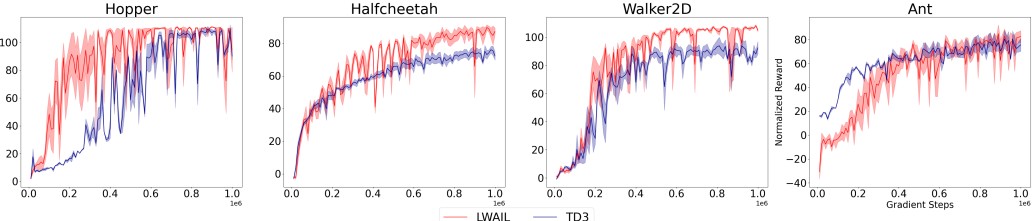

Figure 5: Ablations on the pseudo reward learned by discriminator and ground truth reward. Our learned reward (red) performs equal to or better than TD3 (blue) with ground truth reward.

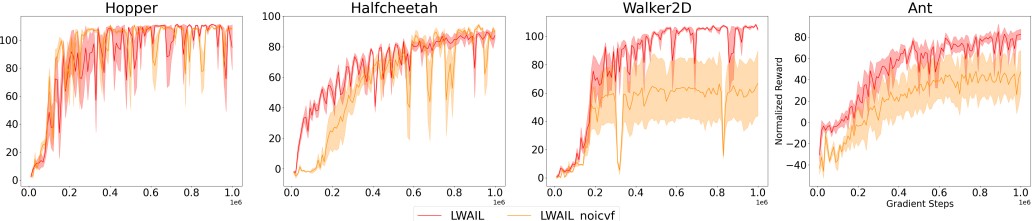

Figure 6: Ablations on the ICVF-learned embedding part, where the red curve is LWAIL and the orange curve is LWAIL without ICVF-learned $\phi$ for $f(\cdot, \cdot)$. In more complicated environments like Walker2d and Ant. The result shows that with ICVF-learned metric, LWAIL performs better.

## 5 RELATED WORK

**Imitation by occupancy matching / adversarial training.** GAIL (Ho & Ermon, 2016) is one of the first works to study adversarial imitation learning. A bi-level optimization of "RL over inverse RL" is considered, which corresponds to state-action occupancy matching between an expert and the learner's policy. Followup works (Fu et al., 2018; Kostrikov et al., 2019; Torabi et al., 2018b; Zhu et al., 2020) have further explored the adversarial training paradigm, jointly training 1) a discriminator to distinguish policy occupancy from expert occupancy; and 2) the policy fitting expert demonstrations. While some occupancy matching works like DIstribution Corrected Estimation (DICE) (Ma et al., 2022; Kim et al., 2022a; Yan et al., 2024) are trained using a single-level optimization instead of an adversarial setup, they essentially derive a closed-form solution for the policy given the discriminator. Most works in this field, however, focuses on $f$-divergence (especially KL (Zhu et al., 2020) or $\chi^2$ (Ma et al., 2022)) minimization. In contrast, our work studies the Wasserstein distance and overcomes initial limitations.

**Wasserstein-based imitation learning.** The Wasserstein distance (Kantorovich, 1939), is widely adopted in IL/RL (Xiao et al., 2019; Agarwal et al., 2021; Fickinger et al., 2022). It provides a geometry-aware measure between policy occupancies. Among different forms of the Wasserstein distance, the primal form (Dadashi et al., 2021; Luo et al., 2023; Yan et al., 2024; Bobrin et al., 2024) and the Rubinstein-Kantorovich dual (Kantorovich & Rubinstein, 1958; Zhang et al., 2020b; Garg et al., 2021; Sun et al., 2021) are most prominent. The former allows for larger freedom in its underlying metric, but requires a regularizer (Yan et al., 2024), surrogates (Dadashi et al., 2021), or a direct match between trajectories (Luo et al., 2023; Bobrin et al., 2024). The latter is easier to optimize with gradient descent, but the underlying metric is limited to Euclidean, which is often suboptimal (Stanczuk et al., 2021; Yan et al., 2024). Our work chooses the latter but overcomes its shortcomings.

Among all these works, IQ-learn (Garg et al., 2021) is most similar to ours. Our objective in Eq. (4) is a special case of IQ-learn with Wasserstein distance. However, three key differences exist: 1) IQ-learn uses SAC (Haarnoja et al., 2018) instead of TD3; 2) IQ-learn focuses on $\chi^2$-divergence in the online setting, which was found to be less effective in several prior works (Ma et al., 2022; Yan et al., 2024); 3) We point out and overcome the underlying metric limitation by adopting ICVF, which is not considered in IQ-learn.

**Imitation from observation.** Imitation (Learning) from Observation (LfO) aims to retrieve an expert policy without labeled actions. This is particularly interesting in robotics, where the expert action can be either inapplicable during cross-embodiment imitation (Sermanet et al., 2017) or unavailable when imitating from videos (Pari et al., 2022). The three primary strategies of LfO can be categorized as follows: 1) minimizing an occupancy divergence through DICE methods (Zhu et al., 2020; Lee et al., 2021; Ma et al., 2022; Kim et al., 2022a;b; Yan et al., 2024) or iterative inverse-RL updates (Torabi et al., 2018b; Xu & Denil, 2019; Zolna et al., 2020); 2) predicting missing actions through inverse dynamics modeling (Torabi et al., 2018a; Kumar et al., 2019); and 3) similarity-based reward assignment (Sermanet et al., 2017; Chen et al., 2019; Wu et al., 2019). Our work belongs to the second category, and adopts the Wasserstein distance as the measure between occupancies, which improves results over prior works.

**Offline-to-online IL.** While offline IL (Zolna et al., 2020; Ma et al., 2022; Kim et al., 2022a) and online IL (Ho & Ermon, 2016; Fu et al., 2018) are both well-studied areas, offline-to-online IL is relatively under-explored, especially when compared with offline-to-online RL (Schmitt et al., 2018; Kostrikov et al., 2022; Zhang et al., 2023) which combines the best of offline RL (high data efficiency) and online RL (active data collection). While there are some works that use offline data to aid online imitation (Watson et al., 2024) by building dynamic models (Chang et al., 2021; Yue et al., 2023) or aligning discriminator and policy (Yue et al., 2024), they have two shortcomings compare to our proposed method: 1) their solution requires medium-to-high quality offline data and does not work well with random offline data, while our ICVF-learned metric works well with random offline data; 2) they require state-action pairs for expert demonstrations, while our method only requires expert states.

**State embedding.** Many works have explored the possibility of learning a good state space embedding that better captures the dynamics of the environment and boosts RL performance (Zhang et al., 2020a; Ghosh et al., 2023; Modi et al., 2024). These works can be roughly categorized into two groups: 1) the 'theoretical group', which focuses on state equivalence (also known as "bisimulation") (Zhang et al., 2020a; Kemertas & Aumentado-Armstrong, 2021; Le Lan et al., 2021) and the low-rank property of the MDP (Agarwal et al., 2020; Uehara et al., 2022; Modi et al., 2024); and 2) the 'empirical group' often tested on visual RL with high-dimensional input (Anand et al., 2019; Laskin et al., 2020; Yarats et al., 2022)), which focuses on representation learning (Ha & Schmidhuber, 2018; Hafner et al., 2023; Bruce et al., 2024), autoencoder methods (Senthilnath et al., 2024), and contrastive learning (Sermanet et al., 2017; Anand et al., 2019; Laskin et al., 2020). The recently proposed ICVF (Ghosh et al., 2023) studies an empirical, intention-based method for state embedding computation. It was shown to be effective in downstream tasks (Ghosh et al., 2023; Bobrin et al., 2024). Our work is the first to leverage ICVF state embeddings to successfully overcome the metric limitation of the KR duality of the Wasserstein distance.

## 6 CONCLUSION

We propose a novel adversarial imitation learning approach for state-only distribution matching using the Wasserstein distance. Unlike prior methods that rely on Euclidean distance metrics, we optimize this distance metric by leveraging an embedding learned by the Intention Conditioned Value Function (ICVF), which captures environmental dynamics. This allows us to better align the state distributions between the expert and the agent, even when only sparse state-only demonstrations are available. Through multiple experiments on the Maze2D and MuJoCo environments, we demonstrate that the ICVF-learned distance metric outperforms several baselines, enabling more efficient and accurate imitation from limited expert data with only one expert trajectory. We believe our work provides a new direction for improving state-only imitation learning by using the Wasserstein distance while addressing the limitations of traditional distance metrics.

**Limitations**. Similar to other prior adversarial imitation learning methods such as WDAIL (Zhang et al., 2020b), our pipeline requires an iterative update of the actor-critic agent and the discriminator during online training. The update frequency needs to be balanced during training. Also, testing our method on more complicated environments, such as image-based ones, is interesting future research.

## 7 REPRODUCIBILITY STATEMENT

We include the procedure of our algorithm in Alg. 1. For environments used in our experiments, we list their details in Sec. 4.1 (for Maze2d) and Appendix B.1 (for MuJoCo); for datasets in our experiment, we list their statistics in Appendix B.2; for the hyperparameters of our method, we list them in Tab. 3 in Appendix B.3; for implementation of the baselines, their related repositories and licenses, we summarize them in Appendix B.4. Finally, we state our computational resource consumption in Appendix E. We will publish our code upon acceptance.

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

APPENDIX: LATENT WASSERSTEIN ADVERSARIAL IMITATION LEARNING

Our appendix is organized as follows. In Sec. A, we discuss ICVF and provide a more detailed explanation of Eq. (6). In Sec. B, we provide the details of the environment in our experiments (Sec. B.1), the dataset used in our experiments (Sec. B.2), the hyperparameters we used for our method (Sec. B.3), and the details for our baselines (Sec. B.4). In Sec. D, we summarize the notation used in our paper. Finally, in Sec. E, we state the computational resource used for running our experiments.

## A   EXTENDED PRELIMINARIES

**Intention Conditioned Value Function (ICVF).** Intuitively, $V(s, s_+, z)$ is designed to evaluate the likelihood of the following question: *How likely am I to see $s_+$ if I act to perform $z$ from state $s$?* The learning of ICVF is similar to other value-learning algorithms. ICVF satisfies the following Bellman equation:

$$V(s, s_+, z) = \mathbb{E}_{a \sim \pi_z^*} \left[ \mathbb{I}(s = s_+) + \gamma \mathbb{E}_{s' \sim P_z(\cdot|s_t)} \left[ V(s', s_+, z) \right] \right],$$
$$\text{where } \pi_z^* = \arg\max_a r_z(s) + \gamma \mathbb{E}_{s'} \left[ V(s', z, z) \right]. \tag{10}$$

Here, $(s, s')$ is a transition and $P_z(s_{t+1}|s)$ is the transition probability from $s_t$ to $s_{t+1}$ when acting according to intent $z$. Further, $r_z$ defines the agent's objective for a particular intention $z$. Note, $r_z(s)$ is not the ground truth reward signal. Instead, it describes whether a state $s$ is desirable by intent $z$ and thus depends on data; in other words, the agent aims to maximize the reward specified by $r_z$ when pursuing intention $z$. The original reward is not needed in ICVF training.

The original paper adopts implicit Q-learning (IQL) for ICVF learning. In one update batch, we sample transition $(s, s')$, potential future outcome $s_+$, and intent $z$. Similar to the original IQL (Kostrikov et al., 2022), we update the critic with asymmetric critic losses to avoid out-of-distribution overestimation. To do this, we apply different weights on critic loss with respect to the positivity of *advantage*. Note, as we care about whether the transition $(s, s')$ corresponds to acting with intention $z$, our goal $s_+$ is equal to $z$. Thus, the advantage $A$ is defined as:

$$A = r_z(s) + \gamma V_\theta(s', z, z) - V_\theta(s, z, z). \tag{11}$$

Following that, the critic loss is defined as:

$$\mathcal{L}(V_\theta) = \mathbb{E}_{(s,s'),z,s_+} \left[ |\alpha - \mathbb{I}(A < 0)|(V_\theta(s, s_+, z) - \mathbb{I}(s = s_+) - \gamma V_{\text{target}}(s', s_+, z))^2 \right]. \tag{12}$$

## B   EXPERIMENTAL DETAILS

### B.1   ENVIRONMENTS

We use five MuJoCo (Todorov et al., 2012) and D4RL (Fu et al., 2020) environments: Maze2d, hopper, halfcheetah, walker2d and ant. The environment specifications for maze2d are provided in Sec. 4.1. In this section, we will briefly introduce the other MuJoCo environments. Fig. 7 provides an illustration of those environments.

1. **Hopper.** The hopper environment (as well as the other three environments) is a locomotion task. In hopper, the agent needs to control a single-legged robot leaping forward in a 2D space with $x$- and $z$-axis. The 11-dimensional state space encompasses joint angles and velocities of the robot, while the 3-dimensional action space corresponds to torques applied on each joint.

2. **Halfcheetah.** In the Halfcheetah environment, the agent needs to control a cheetah-shaped robot to sprint forward. It also operates in a 2D space with $x$- and $z$-axis, but has a 17-dimensional state representing joint positions and velocities, and a 6-dimensional action space that modulates joint torques.

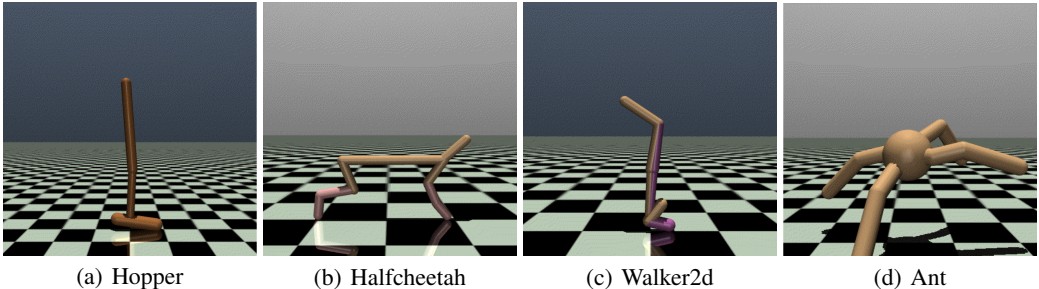

| (a) Hopper | (b) Halfcheetah | (c) Walker2d | (d) Ant |

Figure 7: Illustration of the MuJoCo (Todorov et al., 2012) environments we test in Sec. 4.2.

3. **Walker2d.** As implied by its name, in Walker2d, the agent needs to control a 8-DoF bipedal robot to walk in the two dimensional space. It has a 27-dimensional state space and an 8-dimensional action space.

4. **Ant.** Different from the other three environments, the Ant environment is a 3D setting where the agent navigates a four-legged robotic ant moving towards a particular direction. The state is represented by 111 dimensions, including joint coordinates and velocities, while the action space has 8 dimensions.

## B.2 DATASETS

For expert datasets of the MuJoCo environments, we use 1 trajectory from the D4RL expert dataset, which has 1000 steps. Some baselines such as PWIL (Dadashi et al., 2021) employ a *subsampling* hyperparameter, which creates a low-data training task by taking only one state/state-action pair from every 20 steps of the expert demonstration. For fairness, we set all baselines' subsampling factors to be 1, i.e., no subsampling.

| Dataset | Size | Normalized Reward (Expert is 100) |
|---|---|---|
| Hopper-random-v2 | 999996 | $1.19 \pm 1.16$ |
| HalfCheetah-random-v2 | 1000000 | $0.07 \pm 2.90$ |
| Walker2d-random-v2 | 999997 | $0.01 \pm 0.09$ |
| Ant-random-v2 | 999930 | $6.36 \pm 10.07$ |

Table 2: The basic statistics of the random datasets from D4RL (Fu et al., 2020) applied in our experiments. It is apparent that all these data are of very low quality compared to an expert, yet our ICVF-learned metric still works well.

## B.3 HYPERPARAMETERS

Tab. 3 summarized the hyperparameters for our method. We use the same settings for all environments, and keep hyperparameters identical to TD3 (Fujimoto et al., 2018) and ICVF (Ghosh et al., 2023) whenever possible.

## B.4 BASELINES

We use several different github repositories for our baselines. We use default settings of those repos, except for the number of expert trajectories (which is set to 1) and the subsampling factor (see Appendix B.2). Below are the repos we used in our experiments for each baseline:

- *BC (Ross et al., 2011), GAIL (Ho & Ermon, 2016), AIRL (Fu et al., 2018):* We use the *imitation* (Gleave et al., 2022) library, which provides clean implementations of several imitation learning algorithms and has a MIT license.

- *OPOLO (Zhu et al., 2020), DACfO (Kostrikov et al., 2019), BCO (Torabi et al., 2018a), GAIfO (Torabi et al., 2018b):* We use OPOLO's official code (https://github.com/

| Type | Hyperparameter | Value | Note |
|------|----------------|-------|------|
| ICVF. | Network Size of $\phi$ | [256, 256] | same as original paper |
| Disc. | Network Size | [64, 64] | |
| | Activation Function | ReLU | |
| | Learning Rate | 0.001 | |
| | Update Epoch | 40 steps | |
| | Update interval | 4000 | |
| | Batch Size | 4000 | |
| | Optimizer | Adam | |
| | Gradient Penalty coefficient | 10 | |
| Actor | Network Size | [256, 256] | |
| | Activation Function | ReLU | |
| | Learning Rate | 0.0003 | |
| | Training length | 1M steps | |
| | Batch Size | 256 | |
| | Optimizer | Adam | |
| Critic | Network Size | [256, 256] | |
| | Activation Function | ReLU | |
| | Learning Rate | 0.001 | |
| | Training Length | 1M steps | |
| | Batch Size | 256 | |
| | Optimizer | Adam | |
| | $\gamma$ | 0.99 | discount factor |

Table 3: Summary of the hyperparameters of LWAIL.

| | Hopper | HalfCheetah | Walker | Ant | Average |
|---|--------|-------------|--------|-----|---------|
| 1 trajectory | $110.52 \pm 1.06$ | $86.71 \pm 5.67$ | $105.30 \pm 2.33$ | $80.56 \pm 13.09$ | 95.77 |
| 5 trajectories | $107.65 \pm 7.47$ | $93.28 \pm 1.97$ | $107.32 \pm 1.36$ | $87.23 \pm 10.43$ | 98.87 |
| All expert dataset | $109.34 \pm 3.87$ | $94.18 \pm 3.12$ | $104.37 \pm 1.97$ | $90.81 \pm 9.61$ | 99.67 |

Table 4: Ablation on using multiple trajectories as expert demonstrations. Our method shows consistent expert-level performance regardless of the number of expert demonstrations.

    `illidanlab/opolo-code`), where DACfO, BCO and GAIfO are integrated as baselines, which does not have a license.

- *OLLIE (Yue et al., 2024):* We tried to use the official code but it can't be executed due to non-trivial typos. Thus we use their reported numbers on random dataset instead.
- *PWIL:* We use another widely adopted imitation learning repository (Arulkumaran & Ogawa Lillrank, 2023) (`https://github.com/Kaixhin/imitation-learning`), which has an MIT license.
- *WDAIL:* We use their official code (`https://github.com/mingzhangPHD/Adversarial-Imitation-Learning/tree/master`), which does not have a license.
- *IQlearn:* We use their official code (`https://github.com/Div99/IQ-Learn/tree/main`) with a research-only license.

## C  MORE ABLATIONS

In this section, we provide additional ablation results of our method. We report normalized reward (higher is better) for all results.

### C.1  MULTIPLE TRAJECTORIES

To demonstrate robustness of our method even if the expert data is scarce, we test our method with 5 expert trajectories and the whole expert dataset (1M transitions). Tab. 4 summarizes the results. We observe consistent compelling performance regardless of the number of expert trajectories.

### C.2  EMBEDDINGS

In this section, we compare our method with ICVF embeddings to use of other embeddings. It is worth noting that while there are embedding methods for RL/IL, most of them are not applica-

| | Hopper | HalfCheetah | Walker | Ant | Average |
|---|---|---|---|---|---|
| LWAIL | $110.52 \pm 1.06$ | $86.71 \pm 5.67$ | $105.30 \pm 2.33$ | $80.56 \pm 13.09$ | **95.77** |
| PW-DICE | $110.60 \pm 0.77$ | $46.07 \pm 27.95$ | $106.63 \pm 1.03$ | $85.36 \pm 8.12$ | 87.16 |
| CURL | $105.70 \pm 1.22$ | $87.62 \pm 5.10$ | $102.97 \pm 4.19$ | $52.03 \pm 8.33$ | 87.08 |
| No Embedding | $108.34 \pm 3.42$ | $85.98 \pm 3.42$ | $62.39 \pm 20.43$ | $40.72 \pm 18.95$ | 74.36 |

Table 5: Ablation of different embedding methods with LWAIL. The result shows that ICVF embeddings outperform other contrastive learning-based embeddings.

| | Hopper | HalfCheetah | Walker | Ant | Average |
|---|---|---|---|---|---|
| LWAIL | $110.52 \pm 1.06$ | $86.71 \pm 5.67$ | $105.30 \pm 2.33$ | $80.56 \pm 13.09$ | **95.77** |
| LWAIL_subsample | $109.00 \pm 0.46$ | $86.73 \pm 7.02$ | $106.13 \pm 2.47$ | $83.21 \pm 8.80$ | **96.27** |
| WDAIL_subsample | $108.21 \pm 4.90$ | $35.41 \pm 2.07$ | $114.32 \pm 2.07$ | $83.87 \pm 10.92$ | 85.45 |
| IQlearn_subsample | $60.26 \pm 14.21$ | $4.12 \pm 1.03$ | $8.31 \pm 1.48$ | $5.32 \pm 3.87$ | 19.50 |

Table 6: Ablation on subsampled expert trajectories. The result shows that LWAIL is robust to subsampled expert demonstrations and outperforms other baselines with subsampled expert demonstrations.

ble to our scenario. For instance, most empirical state embedding methods are for visual environments (Meng et al., 2023; Sermanet et al., 2018) or for cross-domain dynamics matching (Duan et al., 2017; Franzmeyer et al., 2022). Among theoretical state embedding methods, low-rank MDPs (Modi et al., 2024) are not applicable to the MuJoCo environment, and bisimulation (Zhang et al., 2020a) requires a reward signal which is not available in imitation learning.

Nonetheless, we identify two contrastive learning-based baselines that are most suitable for our scenario: CURL (Laskin et al., 2020) and PW-DICE (Yan et al., 2024). Both methods use InfoNCE (Oord et al., 2018) as their contrastive loss for better state embeddings. Their difference: 1) CURL updates embeddings with an auxiliary loss during online training, while PW-DICE updates embeddings before all other training; 2) CURL compares the current state with different noises added as positive contrast examples, while PW-DICE uses the next states as positive contrast samples. Tab. 5 summarizes the results. The result shows that 1) state embeddings generally aid learning; and 2) our proposed method works best.

### C.3 SUBSAMPLE

To validate the robustness of our policy, we provide results with subsampled expert trajectories, a widely-adopted scenario in many prior works such as PWIL and IQ-learn. Only a small portion of the complete expert trajectories are present. Our subsample ratio is 10, i.e., we take 1 expert state pair out of adjacent 10 pairs. Tab. 6 summarizes the results, which show that 1) our method with subsampled trajectories outperforms Wasserstein-based baselines such as WDAIL (Zhang et al., 2020b) and IQlearn (Garg et al., 2021), and 2) the performance of our method is not affected by incomplete expert trajectories.

### C.4 DOWNSTREAM RL ALGORITHM

We used TD3 as our downstream RL algorithm rather than PPO with entropy regularizer. Our choice is motivated by better efficiency and stability, especially because TD3 is an off-policy algorithm which is more robust to the shift of the reward function and our adversarial training pipeline. We ablate this choice of the downstream RL algorithm and show that TD3 outperforms PPO in our framework. Tab. 7 summarizes the results.

### C.5 ICVF EMBEDDING WITH OTHER METHODS

We also show that our proposed solution outperforms existing methods with ICVF embedding, both Wasserstein-based (IQlearn, WDAIL) and $f$-divergence based. The results are summarized in Tab. 8 (using average reward; higher is better). We find that 1) our method outperforms prior methods with ICVF embedding, and 2) ICVF does not necessarily improve the performance of prior methods,

| | Hopper | HalfCheetah | Walker | Ant | Average |
|---|---|---|---|---|---|
| LWAIL+TD3 (original) | $110.52 \pm 1.06$ | $86.71 \pm 5.67$ | $105.30 \pm 2.33$ | $80.56 \pm 13.09$ | **95.77** |
| LWAIL+PPO | $65.21 \pm 4.81$ | $1.02 \pm 0.21$ | $24.13 \pm 2.14$ | $9.12 \pm 0.85$ | 24.87 |

Table 7: Ablation on downstream RL algorithms. The result shows that TD3 works much better than PPO.

| | Hopper | HalfCheetah | Walker | Ant | Average |
|---|---|---|---|---|---|
| LWAIL | $110.52 \pm 1.06$ | $86.71 \pm 5.67$ | $105.30 \pm 2.33$ | $80.56 \pm 13.09$ | **95.77** |
| WDAIL+ICVF | $110.02 \pm 0.53$ | $30.07 \pm 2.32$ | $68.68 \pm 9.16$ | $3.42 \pm 1.01$ | 53.04 |
| IQlearn+ICVF | $29.80 \pm 10.12$ | $3.82 \pm 0.98$ | $6.54 \pm 1.23$ | $8.91 \pm 0.45$ | 12.27 |
| GAIL+ICVF | $8.96 \pm 2.09$ | $0.12 \pm 0.40$ | $3.98 \pm 1.41$ | $-3.09 \pm 0.85$ | 2.49 |

Table 8: ICVF with other methods. Our method far outperforms other methods with ICVF embeddings.

due to other components of our method (e.g., normalized input for the Wasserstein discriminator, downstream RL algorithm).

## C.6 MISMATCHED DYNAMICS

It is worth noting that the very motivation of LWAIL is to find a latent space which aligns well with the environment's true dynamics. Despite this, we agree that there might be cases where the latent space employed in LWAIL does not align with the true dynamics due to inaccurate data, e.g., mismatched dynamics between expert demonstrations and the actual environment. To test such cases, we use the halfcheetah mismatched experts scenario analyzed in SMODICE (Ma et al., 2022): for expert demonstration, the torso of the cheetah agent is halved in length, thus causing inaccurate alignment. We compared our methods with the results reported in the SMODICE paper. Tab. 9 summarizes the final average normalized reward (higher is better). Results show that 1) our method works better than several baselines including SMODICE; and 2) our method is robust to mismatched dynamics.

## C.7 SIGMOID REWARD MAPPING

We adopt the sigmoid function to regulate the output of our neural networks for better stability (similar to WDAIL (Zhang et al., 2020b)). However, one cannot naively apply the sigmoid to the reward function for better performance. To show this, we compare to TD3 with a sigmoid function applied to the ground truth reward. The result is illustrated in Tab. 10. The result shows that a naive sigmoid mapping of the reward does not improve TD3 results.

## C.8 PSEUDO-REWARD METRIC CURVE

To validate the effect of using sigmoid and ICVF embedding for our pseudo-reward generated by $f$, we conduct two experiments:

1) Run a standard setting of LWAIL, and compare pseudo-rewards generated by $f$ with the sigmoid function, and pseudo-rewards without the sigmoid function for the MuJoCo environments. This is illustrated in Fig. 8.

2) Run standard LWAIL and LWAIL without ICVF embedding, and compare pseudo-rewards (with the sigmoid function) for the MuJoCo environments. This is illustrated in Fig. 9.

The result clearly shows that both ICVF-embedding and sigmoid function are very important for pseudo-reward stability and positive correlation with ground-truth reward.

## D LIST OF NOTATIONS

Tab. 11 summarizes the symbols which appear in our paper.

| | Normalized Reward |
|---|---|
| LWAIL | **24.31 ± 4.51** |
| SMODICE | **23.2 ± 7.43** |
| SAIL | 0 ± 0 |
| ORIL | 2.47 ± 0.32 |

Table 9: Performance on the Halfcheetah environment with mismatched dynamics. Our method outperforms baselines.

| Environment | **Hopper** | **HalfCheetah** | **Walker** | **Ant** | **Maze2D** | **Average** |
|---|---|---|---|---|---|---|
| TD3 | 105.54 | 76.13 | 89.68 | 89.21 | 120.14 | 96.14 |
| TD3+Sigmoid reward | 84.23 | 30.76 | 42.55 | 34.79 | 119.03 | 62.27 |

Table 10: Results of TD3 with and without sigmoid applied on the ground truth reward. The results show that applying the sigmoid function does not yield better performance.

# E COMPUTATIONAL RESOURCES

All our experiments are performed with an Ubuntu 20.04 server, which has 128 AMD EPYC 7543 32-Core Processor and a single NVIDIA RTX A6000 GPU. With these resources, our method needs about $65 - 75$ minutes for the MuJoCo environments.

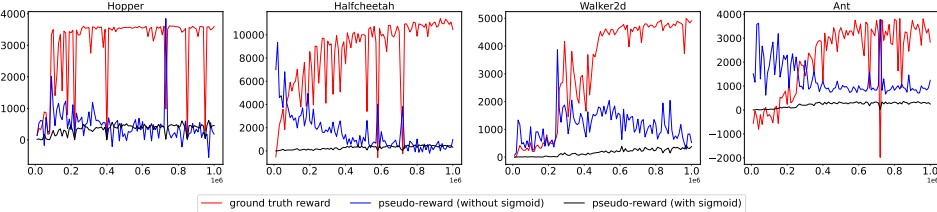

Figure 8: The reward curves of pseudo- and ground-truth reward in a single training session, where pseudo-reward is generated by $f$ following Alg. 1 and serves as the reward signal for our downstream TD3. We note that the pseudo-reward is much more stable and positively correlated with ground-truth reward when using a sigmoid function.

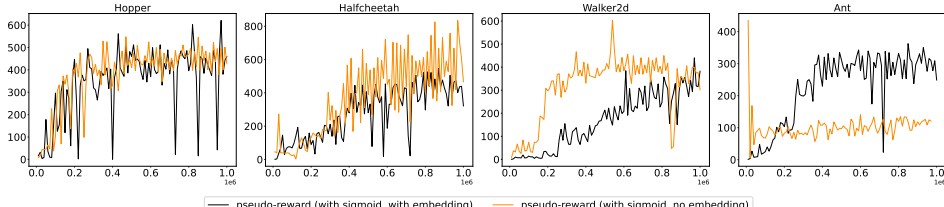

Figure 9: The pseudo-reward curves with and without ICVF embedding in a single training session. We note that without ICVF, the pseudo-reward is generally less stable (e.g. fluctuation in halfcheetah and sudden drop in walker2d and ant) and sometimes less correlated with ground-truth reward (e.g. ant environment).

| Name | Meaning | Note |
|---|---|---|
| $\mathcal{S}$ | State space | |
| $s$ | State | $s \in \mathcal{S}$ |
| $\mathcal{A}$ | Action space | |
| $a$ | Action | $a \in \mathcal{A}$ |
| $t$ | Time step | $t \in \{0, 1, 2, \dots\}$ |
| $\gamma$ | Discount factor | $\gamma \in$ $[0, 1)$ |
| $r$ | Reward function | $r(s, a)$ for single state-action pair |
| $P$ | Transition | $P(s'\|s, a) \in \Delta(\mathcal{S})$ |
| $E$ | Expert dataset | state-only expert demonstrations |
| $I$ | Random dataset | state-action trajectories of very low quality |
| $\pi$ | Learner policy | The policy we aim to optimize |
| $d_s^\pi$ | State occupancy of $\pi$ | $d_s^\pi(s) = (1 - \gamma) \sum_{i=0}^{\infty} \gamma^i \Pr(s_i = s)$, where $s_i$ is the $i$-th state in a trajectory |
| $d_{ss}^\pi$ | State-pair occupancy of $\pi$ | $d_s^\pi(s, s') = (1 - \gamma) \sum_{i=0}^{\infty} \gamma^i \Pr(s_i = s, s_{i+1} = s')$, where $s_i$ is the $i$-th state in a trajectory |
| $d_{ss}^E$ | State-pair occupancy of expert policy | The expert policy here is empirically induced from $E$ |
| $c$ | Underlying metric for Wasserstein distance | |
| $f$ | Dual function / Discriminator | Dual function in Rubinstein dual form of 1-Wasserstein distance; also a discriminator from adversarial perspective and a reward model from IRL perspective |
| $\Pi$ | Wasserstein matching variable | In our case, $\sum_{s \in \mathcal{S}} \Pi(s, s') = d_s^E(s')$, $\sum_{s' \in \mathcal{S}} \Pi(s, s') = d_s^\pi(s)$ |
| $\mathcal{W}_1$ | 1-Wasserstein distance | |
| $s_+$ | Outcome state | |
| $z$ | Latent intention | |
| $V$ | Value function | takes $s, s_+, z$ as input in ICVF; only takes $s$ in normal RL |
| $V_{\text{target}}$ | Target value | target value function in the critic objective of RL |
| $\mathbb{I}$ | indicator function | $\mathbb{I}[\text{condition}] = 1$ if the condition is true, and $= 0$ otherwise |
| $\phi$ | State representation (embedding) | the embedding function we use for $f$; $\phi(s) \in \mathbb{R}^d$ |
| $T$ | Counterfactual intention | $T(z) \in \mathbb{R}^{d \times d}$ |
| $\psi$ | Outcome representation | $\psi(s_+) \in \mathbb{R}^d$ |
| $\alpha$ | ICVF constant | $\alpha \in (0.5, 1]$ |
| $\sigma$ | Sigmoid function | |

Table 11: A list of symbols used in the paper. The first part focuses on RL-specific symbols. The second part details Wasserstein-specific notation. The third part summarizes ICVF-specific symbols (Sec. 3.2).

