# OpenReview forum: "Latent Wasserstein Adversarial Imitation Learning"
_ICLR.cc/2025/Conference — Submitted to ICLR 2025_

### Official Review · Reviewer_Eovk · 2024-10-27

**Soundness:** 3
**Presentation:** 1
**Contribution:** 2
**Rating:** 5
**Confidence:** 4

**Summary:**

The paper studies the distribution matching idea in imitation learning and considers formulating this idea in a latent space learned through Intention Conditioned Value Function representation. The reward function is then defined in such latent space with a sigmoid transformation to turn it into non-negative and bounded values. The paper demonstrates that such idea can yield better representation of trajectories, which consequently lead to better empirical performance.

**Strengths:**

* [Originality] The paper proposes to study the distribution matching in a latent space, learned from Intention Conditioned Value Function method, and argues that the latent space can yield a better metric for the distribution matching under Wasserstein distance. This can be inspiring to imitation learning and reinforcement learning
* [Quality and clarity] Overall, the intro and related work are well written, showing a good understanding of the main challenge in imitation learning. The visualization of latent space representation and illustration of a better metric also greatly help readers to understand the paper.
* [Significance] Considering imitation learning in a well-structured latent space can be more effective than the original trajectory space. The paper can contribute positively to the development of such direction.

**Weaknesses:**

### Some vague, inconsistent and confusing descriptions of the method
1. “clip threshold $c_0 > 0$ is added to the target action $a’$”. But the following equation applies clip to the random noise. Should it be $clip(a’)$ or $clip(\epsilon)$?
 2. “the metric $c(s, s’)$ in Eq. (4) is inherently limited to be Euclidean”. There is no $c(s, s’)$ in Eq. (4). Here the metric refers to the 1-Lipschitz constraint?
3. the reward used by TD3 algorithm is defined as $r(s, s’) = \sigma(-f(s, s’))$ (as described in line 304 and also specified in Line 7 in Algo 1). Why is the latent space not used for reward function? And how does the function $f$ takes $s$ and $s’$ as input to predict the reward while at the same time takes $\phi(s’)$ and $\phi(s)$ (vectors of different dimensions) as input. The reward definition is not even consistent with the optimization in Eq. (9).
4. some notations are not clear: the summation index t in Eq. (5) starts from 0 and goes to infinity? How to ensure that it is well defined for $\gamma=1$ since $\gamma \in [0, 1]$; what’s the minimization in Eq. (6) over? Also, can authors explain the $\alpha$ in Eq. (6) and why it needs to be set in between 0.5 and 1?
5. “Euclidean distance … in latent space … capturing the structure of the environment more faithfully”. I don’t know how to interpret this on Fig 2. What’s the structure of the Hopper and HalfCheetah? Also, why is Euclidean distance in this space a more suitable metric? From my understanding of Fig 2, the traj in original space has a better coverage of state values than the traj in the latent space. So the former representation would be better for distribution matching learning?

### Vagueness in the experiment and lack of important ablations
1. Re the experimental results in Fig 3,  is the reward plotted corresponding to $-f(s, s’)$ or $\sigma(-f(s, s’))$? Why distinctive reward signal is better? It looks from Fig 3b that most rewards in latent space is close to 0 though
2. The reward in LWAIL is defined as the $\sigma(-f)$, which is naturally denser than the original reward function in Maze2D. The paper should compare the performance of TD3 with sigmoid transformation of the spare reward.
3. In line 416, re “our original method”, do authors mean $f(s, s’)$ or $f(s, s’-s)$?. Further, re “agents without … embedding tend to remain in stable but relatively low-reward states, exhibiting less tendency to engage in riskier explorations”, can authors explain how this phenomenon is related to the ICVF method adopted by the paper?
4. The paper should provide more ablation studies on the sigmoid transformation of reward function and the embedding difference as input, i.e., $\phi(s’) - \phi(s)$ rather than $\phi(s’)$. It is widely known that in Mujoco control benchmarks transforming rewards to non-negative will have a positive influence on the RL performance. Particularly, in Fig 5, the pseudo reward is always positive as it is the output of a sigmoid function while the ground truth reward may not be. Additionally, a pseudo reward generated by $f(s, s’-s)$ (i.e., without the embedding) should also be reported. Further, in Fig 6, it is unclear what it means for LWAIL without ICVF-learned $\phi$. The paper should explain explicitly how the LWAIL is trained without ICVF and should also report the results on using $f(s, s’-s)$.

**Overall I found the empirical evaluation appears not quite convincing in its current version and many important details are missing. The inconsistent descriptions in the method part further makes the experiment part harder to understand.**

**Questions:**

1. what does it mean by “policy’s understanding of state transitions”?
2. the paper introduces the Kantorovich-Rubinstein duality and then just refers this duality form as Rubinstein dual. Wouldn’t it more accurate to refer it as Kantorovich-Rubinstein duality or KR duality?
3. why is the reward function in Line 7 Algo 1 computed from $f(s, s’)$? Not $f(\phi(s), \phi(s’) - \phi(s))$?
4. how is the representation network $\phi$ being used in policy update? There is no mention of $\phi$ from line 5 – 12 in Algo 1.
5. I found it very hard to grasp the main idea in Section Overcoming Metric Limitations, both intuition-wise and methodology-wise. It would be great if this part can be improved.

---

> ### Author Response · Authors · 2024-11-22
> **Response to Reviewer Eovk (Part 1 of 2)**
>
> Thanks for your constructive advice. We answer questions next:
>
> **Q1. Notation.**
>
> Thank you for carefully reading our paper. We have fixed the notation in our revised paper.
>
> **1. clip threshold (description weakness 1).** The correct understanding (Gaussian noise $\epsilon$ with variance $\sigma^2>0$ and clip threshold $c_0>0$) is added to the target action $a'$. We have modified the submission to clarify this.
>
> **2. There is no $c(s,s’)$ in Eq. 4 (description weakness 2).** The $c(s,s’)$ is hidden in the constraint $\|\|f\|\|_L\leq 1$, as explained in our explanation to Eq. (1) (which is essentially $\frac{|f(s)-f(s')|}{c(s, s')}\leq 1$). To clarify this, we have updated Sec. 3.2.
>
>
> **3. The input for $f$ (description weakness 3; questions 3).** All $f(\cdot, \cdot)$ in Sec. 3.3 and later use $f(\phi(s),\phi(s')-\phi(s))$ as input. Thus there is no inconsistency in our ablation. The reward function is using latent space. We have updated Sec. 3.3 to reflect this.
>
>
> **4. The summation index t in Eq. 5 and definition of $gamma$ (description weakness 4).** In RL, $\gamma=1$ is usually used in finite-horizon MDPs with total reward being the sum over each timestep, while $\gamma<1$ is usually used in infinite-horizon MDPs with reward summing over infinite timesteps in the future using a decaying weight. We adopt the infinite horizon MDP framework. We have updated $\gamma$ to $[0, 1)$ in our preliminary section.
>
> **5. Explanation of $\alpha$ (description weakness 4).** $\alpha$ is an important parameter in IQL. As IQL is an algorithm that penalizes over-optimistic evaluation for out-of-distribution data in offline RL, it uses an asymmetric critic objective where overestimation is punished harder than underestimation. Thus, we have $\alpha>0.5$, such that $| \alpha - \mathbb{I}(A < 0) |$ has greater values for advantage $A>0$.
>
> **6. What is Eq. 6 Minimizing (description weakness 4)?** Thanks for pointing this out. The minimization is over the value function $V_\theta$ and we have updated Eq. (6) in our paper. The detailed objective is listed in Appendix A.
>
> **7. KR duality instead of Rubinstein duality. (questions 2)** Thanks for pointing this out. We have updated Rubinstein duality in our paper to KR duality.
>
> **Q2. Why distinctive reward signals are better in Fig. 3b, as most rewards in latent space are close to 0 though, and why less tendency for riskier explorations is related to ICVF behavior? (experiment weakness 1 and 3)**
>
> The plotted reward corresponds to $\sigma(-f(\phi(s),\phi(s’)-\phi(s)))$. It is shown in Fig. 3b that there exist several trails of trajectories around the goal, which is an evidence that the latent space is more aware of the environmental dynamics recovered from the trajectories. While the advantage might not be that obvious for a simple 2D environment (note our ablation in Fig. 3c still shows that our method outperforms TD3 with ground truth reward), such awareness of trajectory dynamics can be much more informative for agents as illustrated in our updated Fig. 2. Without the embeddings, the chance that exploration of online inverse RL falls into low-reward areas increases, thus decreasing the value estimation of adjacent states which in turn harms exploration.
>
> **Q3.  The reward in LWAIL is naturally denser than the original reward function in Maze2D. The paper should compare the performance of TD3 with sigmoid transformation of the spare reward (experiment weakness 2).**
>
> We adopt the sigmoid function to regulate the output of our neural networks for better stability. Use of this was shown in WDAIL [1]. Importantly, the sigmoid function is not related to a density, but rather a technique to stabilize training.
>
> To further address the reviewer’s question, we ablated TD3 with a sigmoid function on the ground truth reward. The result is illustrated below. The result shows that a naive sigmoid mapping on the reward function does not help TD3.
>
> Env | Hopper | HalfCheetah | Walker | Ant | Average
> ---|---|---|---|---|---
> TD3 | 105.54 ($\pm$1.48) | 76.13 ($\pm$4.98) | 89.68 ($\pm$3.21) | 89.21 ($\pm$2.86) | **90.14**
> TD3 with sigmoid | 84.23 ($\pm$3.44)| 30.76 ($\pm$9.21) | 42.55 ($\pm$6.28) | 34.79 ($\pm$5.02)| 48.08

---

> > ### Comment · Reviewer_Eovk · 2024-11-27
> > **Thank you for the response**
> >
> > Rethe sigmoid transformation, thanks for providing the results on Mujoco controls. How about the Maze2D?

---

> ### Author Response · Authors · 2024-11-22
> **Response to Reviewer Eovk (Part 2 of 2)**
>
> **Q4. the input of network and ablation study. (experimental weakness 4)**
>
> We use $\phi(s’) - \phi(s)$ rather than $\phi(s)$ to account for the state-transition effect of the action. As $s$ can reach $s’$ in the dataset within one step, the dynamic-aware embeddings $\phi(s)$ and $\phi(s’)$ are often close; by using $\phi(s’) - \phi(s)$, the network can focus on learning the difference between embeddings. Preliminary experiments showed that this improves training stability. We already included the ablation study of $f(s,s’-s)$ without ICVF embedding in Sec. 4.3 of our paper, illustrated in Fig. 6. We found that without ICVF, the method works generally worse. Also, for ablations to other contrastive learning-based embedding methods, see our updated Appendix C.2.
>
> Env | Hopper | HalfCheetah | Walker | Ant | Average
> ---|---|---|---|---|---
> LWAIL | 110.52 ($\pm$1.06) | 86.71 ($\pm$5.67) | 105.30 ($\pm$2.33) | 80.56 ($\pm$13.09) | **95.77**
> No Embedding | 108.34 ($\pm$3.42) | 85.98 ($\pm$3.42) | 62.39 ($\pm$20.43) | 40.72 ($\pm$18.95) | 74.36
>
> **Q5. What does it mean by “policy’s understanding of state transitions” (questions 1)?**
>
> It means the policy is easier to learn from a state-transition-aware embedding space. If we view the embedding function as a part of the policy, then the policy is more “aware” of the state transitions dynamics when using the embedding.
>
>
> **Q6. How is the representation network $\phi$ being used in policy updates? (questions 4)**
>
> Training $\phi$ is a part of the offline training process. $\phi$ is frozen during online training. This is illustrated in Fig. 1.
>
>
> **Q7. I found it very hard to grasp the main idea in Section Overcoming Metric Limitations, both intuition-wise and methodology-wise. It would be great if this part can be improved (questions 5).**
>
>
> To improve understanding, we have updated the notation and explained the design of the rewards. We have also modified the illustrations and explanations of the latent space in our updated Fig. 2. Also, we included the details for ICVF in our Appendix A due to page limits.
>
> Here we present the intuition of this section again briefly: The denominator of the 1-Lipschitz constraint forces existing Kantorovich-Rubinstein (KR) duality-based methods to use Euclidean distance in practice due to the gradient regularizer. However, the Euclidean distance is typically not environment dynamics-aware (see illustration in Fig. 1a). Hence, intuitively, we expected an embedding space in which the Euclidean distance is dynamics-aware to yield better performance. We find the ICVF embedding space to satisfy this property because the ICVF training process is inherently dynamics-aware. Indeed, our developed method to incorporate ICVF embeddings shows better performance.
>
> **References**
>
> [1] Z. Zhang et al. Wasserstein Distance Guided Adversarial Imitation Learning with Reward Shape Exploration. In DDCLS, 2020.

---

> > ### Comment · Reviewer_Eovk · 2024-11-27
> > **Ablation without ICVF embedding**
> >
> > Thanks for providing more empirical results. In my original review comments, I said "a pseudo reward generated by
> >  (i.e., without the embedding) should also be reported". Is it possible to report this reward? I'm curious to know how the reward would be like? also, is the sigmoid function being applied to transform the reward?

---

> > > ### Author Response · Authors · 2024-11-28
> > >
> > > Thanks for your reply and constructive feedback for our paper. We hope our rebuttal has already addressed most of your questions. Below are our responses to follow-up questions:
> > >
> > > **Q1. Sigmoid transformation on Maze2D.**
> > >
> > > Maze2D is a straightforward environment with a simple reward structure. Due to this simplicity, applying a sigmoid function does not significantly affect the results. The performance comparison using normalized reward (higher is better) is as follows:
> > >
> > > Env | Maze2D
> > > ---|---
> > > TD3 | 120.14 ($\pm$1.31)
> > > TD3 with sigmoid | 119.03 ($\pm$0.95)
> > >
> > > In all pseudo-reward generation processes, we apply a sigmoid function to ensure stability. We added this result to the revised Appendix C.7 as Tab. 10.
> > >
> > >
> > > **Q2. What are the pseudo-reward like?**
> > >
> > > To answer this question, we conduct two ablations on pseudo-reward curves:
> > >
> > > 1) The updated Fig. 8 compares pseudo-rewards generated by $f$ with and without the sigmoid function for the MuJoCo environments. All curves in Fig. 8 are generated with the standard setting of LWAIL.
> > >
> > > 2) The updated Fig. 9 compares pseudo-rewards for the MuJoCo environments under standard LWAIL setting and LWAIL without ICVF. For corresponding ground truth rewards, see Fig. 6 as reference.
> > >
> > > We have updated our result in current Appendix C.8. The result clearly shows that both ICVF-embedding and sigmoid function are crucial for pseudo-reward’s stability and positive correlation with ground-truth reward.
> > >
> > >
> > > **Q3. is the sigmoid function being applied to transform the reward on the results without ICVF embedding?**
> > >
> > > Yes, the sigmoid function is always applied in the ablation in our Fig. 6 and in the updated Appendix C.2 which studies embeddings.

---

> ### Author Response · Authors · 2024-11-25
>
> Dear reviewer Eovk,
>
> Thanks again for your constructive review to improve our paper. As the author-reviewer discussion period is close to its end, we kindly invite you to further consider our response, which we believe has addressed all your concerns raised in the rebuttal. Thank you very much!

---

> ### Author Response · Authors · 2024-12-02
>
> Dear reviewer Eovk,
>
> Thanks again for your effort in providing constructive feedback on improving our paper! As the discussion period will come to an end in less than 24 hours, we would like to know if you have any remaining concern, so that we can address them. We are sincerely looking forward to hearing from you, and are always happy to further discuss with you.

---

### Official Review · Reviewer_hrKU · 2024-11-03

**Soundness:** 3
**Presentation:** 3
**Contribution:** 2
**Rating:** 5
**Confidence:** 4

**Summary:**

The authors present Latent Wasserstein Adversarial Imitation Learning (LWAIL), that uses state-only expert demonstrations and a Wasserstein distance metric computed in a latent space. To achieve this, the method includes a pre-training stage using an Intention Conditioned Value Function (ICVF), which establishes a meaningful latent space. By only requiring a single or limited number of expert state-only trajectories, LWAIL demonstrates competitive performance in imitation learning tasks, as shown in experiments across MuJoCo environments.

**Strengths:**

- Well-written paper with clear objectives.
- The method is simple to implement and has the potential to be easily applied to various existing GAIL methods.
- The problem and solution are well-defined, and the use of ICVF for pre-training is logically explained, highlighting how it aids in capturing the dynamics of state transitions.

**Weaknesses:**

- The paper lacks novelty. Both ICVF and WGAIL are existing methods, and state-only imitation learning is not a new topic.
- The authors' contribution lies in combining these two approaches and conducting imitation learning in the latent state space, claiming that it achieves a more robust policy with fewer samples. However, robots in MuJoCo typically exhibit cyclical behavior, and GAIL-based methods generally require only a small number of episodes. Also, the more robust policy has not been validated with specific results.
- The motivation for performing imitation learning in the latent space is insufficiently explained.
- The implementation codes are not provided.
- Lacks a discussion on the limitations of the method.

**Questions:**

- The tasks in the MuJoCo simulation environment are relatively simple, as they only use state vectors as input. To highlight the necessity of the latent space, would more complex visual imitation tasks be more appropriate? Could you provide related experiments?
- In practice, existing GAIL-based methods could also leverage ICVF to learn in the latent space. Could you provide additional experiments to offer more horizontal comparisons?
- Have the authors explored applying LWAIL in environments with multi-modal state distributions?
- How does LWAIL handle situations where the ICVF-pretrained latent space does not align well with the environment’s true dynamics?

---

> ### Author Response · Authors · 2024-11-22
> **Response to Reviewer hrKU (Part 1 of 2)**
>
> Thanks for your constructive advice. We answer questions next:
>
> **Q1. ICVF and WGAIL are existing methods, and state-only imitation learning is not a new topic.**
>
> 1. While we agree that ICVF and WGAIL are existing methods, this does not imply a lack of novelty for our method. Our developed Wasserstein imitation learning method with ICVF is based on an **important, novel insight**, overlooked in prior works which employ the Kantorovich-Rubinstein (KR) dual. The insight is that ICVF embeddings create an embedded state space where the Euclidean distance is more aligned with dynamic differences between states, thus fixing the core problem of prior Wasserstein imitation learning works with KR duality: reliance on the Euclidean distance between states. To our best knowledge, nobody has tried to address this problem.
>
> 2. It is true that state-only imitation learning is not a new topic, but this again does not imply a lack of novelty. In contrast, prior work in this field shows that we are focusing on a popular and important setting in imitation learning.
>
> 3. We have compared our method to a variety of baselines, and showed that our method outperforms prior works. The results show that our proposed solution successfully benefits from the distance metric which others have not identified. We think this is a valuable contribution for our community.
>
> **Q2. The more robust policy has not been validated with specific results.**
>
> To validate robustness of our policy, we provide results with **subsampled** expert trajectories, a widely-adopted scenario in many prior works such as PWIL and IQlearn. Only a small portion of the complete expert trajectories are present. Our subsample ratio is 10, i.e., we take 1 expert state pair out of adjacent 10 pairs. We use 10 expert trajectories as demonstrations.
>
> Env | Hopper | HalfCheetah | Walker | Ant | Average
> ---|---|---|---|---|---
> LWAIL| 110.52 ($\pm$1.06) | 86.71 ($\pm$5.67) | 105.30 ($\pm$2.33) | 80.56 ($\pm$13.09) | **95.77**
> LWAIL_subsample | 109.00($\pm$0.46) | 86.73 ($\pm$7.02) | 106.13 ($\pm$2.47) | 83.21 ($\pm$8.80) | **96.27**
> WDAIL_subsample | 108.21 ($\pm$4.90) | 35.41 ($\pm$2.07) | 114.32 ($\pm$2.07) | 83.87 ($\pm$10.92) | 85.45
> IQlearn_subsample | 60.26 ($\pm$14.21) | 4.12 ($\pm$1.03) | 8.31 ($\pm$1.48) | 5.32($\pm$ 3.87) | 19.50
>
> We observe that our method can deal with highly incomplete trajectories, underlining its robustness. We also conduct experiments showing the robustness of our method when the dynamics between expert demonstrations and the actual environment don’t match. Results are presented in Q9 of our response.
>
>
> **Q3. What’s the motivation for performing imitation learning in the latent space?**
>
> Our method is an imitation learning method which uses state distribution matching to bring the agent and the expert close. However, as shown in Fig. 2, states close according to the Euclidean metric are not always close in the actual state space. This leads to suboptimal matching results. Worse still, the Euclidean metric is an inherent part of all KR duality-based methods (explained in the preliminary section), which is taken for granted in prior works.
>
> We propose to address this issue by introducing a latent space, where states cluster in a dynamic-aware manner. This makes the notion of “being close” more informative. We have provided an ablation study with and without ICVF embedding in our paper (Fig. 6 in Sec. 4.3). We reproduce the results in the tables below (the metric is normalized reward; higher is better). See our updated Appendix C.2 for ablations to other contrastive learning-based state embedding methods, where our method also improves.
>
> Env | Hopper | HalfCheetah | Walker | Ant | Average
> ---|---|---|---|---|---
> LWAIL | 110.52 ($\pm$1.06) | 86.71 ($\pm$5.67) | 105.30 ($\pm$2.33) | 80.56 ($\pm$13.09) | **95.77**
> No Embedding | 108.34 ($\pm$3.42) | 85.98 ($\pm$3.42) | 62.39 ($\pm$20.43) | 40.72 ($\pm$18.95) | 74.36
>
>
> **Q4. Code is missing.**
> Thanks for pointing this out. We will provide our code upon acceptance. We have updated the reproducibility statement accordingly.
>
> **Q5. discussion on the limitations**
> Thanks for pointing this out. We have updated the submission to include this discussion. We post the limitations here for convenience:
>
> **Limitations.** Similar to other prior adversarial imitation learning methods such as WDAIL, our pipeline requires an iterative update of the actor-critic agent and the discriminator during online training. The update frequency needs to be balanced during training. Also, testing our method on more complicated environments, such as image-based ones, is an interesting avenue for future research.

---

> ### Author Response · Authors · 2024-11-22
> **Response to Reviewer hrKU (Part 2 of 2)**
>
> **Q6. More complex visual environments.**
>
> We select vector-based environments for our evaluation as the quality of the embeddings in visual environments largely depends on visual features extracted from frames (which leads to success for methods like DrQ [1] and RAD [2]) instead of a dynamic-aware property. We think this entangles contributions from dynamics-aware embeddings and visual features, making it harder to clearly assess the contribution.
>
> It is also worth noting that visual environments are not a standard in prior works. For example, BCO, WDAIL, OPOLO and DACfO all do not assess performance on visual environments.
>
> This being said, it is valuable future work to scale to visual environments.
>
> **Q7. Comparison of leveraging ICVF with existing GAIL-based methods.**
>
> It is worth noting that the classic GAIL method aims to minimize the $f$-divergence between the generated state occupancy and the expert occupancy. This ignores the underlying distance between the states. Therefore, these methods do not fit our motivation, which is to employ an embedding where Euclidean distance aligns with dynamic-aware differences.
>
> To show that our proposed solution outperforms existing GAIL-based methods with ICVF embedding, both Wasserstein-based (IQlearn, WDAIL) and $f$-divergence based, we have conducted an experiment and listed the result below (in average reward; higher is better). We found that 1) our method outperforms existing methods with ICVF embedding, and 2) ICVF does not necessarily improve the performance of existing methods, probably due to other components of our method (e.g., downstream RL algorithm as TD3 is more robust in an adversarial framework; see our updated Appendix C.4 for ablations).
>
> Env | Hopper | HalfCheetah | Walker | Ant | Average
> ---|---|---|---|---|---
> LWAIL | 110.52 ($\pm$1.06) | 86.71 ($\pm$5.67) | 105.30 ($\pm$2.33) | 80.56 ($\pm$13.09) | **95.77**
> WDAIL+ICVF | 110.02 ($\pm$0.53) | 30.07 ($\pm$2.32) | 68.68 ($\pm$9.16) | 3.42 ($\pm$1.01) | 53.04
> IQlearn+ICVF | 29.80 ($\pm$10.12) | 3.82($\pm$0.98) | 6.54 ($\pm$1.23) | 8.91 ($\pm$0.45) | 12.27
> GAIL+ICVF | 8.96 ($\pm$2.09) | 0.12($\pm$0.40) | 3.98 ($\pm$1.41) | -3.09 ($\pm$0.85) | 2.49
>
> **Q8. Have the authors explored applying LWAIL in environments with multi-modal state distributions?**
>
> Yes, this is shown in our updated Fig. 2. For instance, in the halfcheetah and ant environment we observe that there exists clusters of states in the embedding space, and the agent transits quickly between the clusters. This implies a multimodal state distribution.
>
> **Q9. How does LWAIL handle situations where the ICVF-pretrained latent space does not align well with the environment’s true dynamics?**
>
> It is worth noting that the very motivation of LWAIL is to find a latent space which aligns well with the environment’s true dynamics. Despite this, we agree that there might be cases where the latent space employed in LWAIL does not align with the true dynamics due to inaccurate data, e.g., mismatched dynamics between expert demonstrations and the actual environment. To test such cases, we use the halfcheetah mismatched experts scenario analyzed in SMODICE [3]: for expert demonstration, the torso of the cheetah agent is halved in length, thus causing inaccurate alignment. We compared our methods with the results reported in the SMODICE paper. Below are the final average normalized rewards (higher is better):
>
> Method | LWAIL | SMODICE | SAIL [4] | ORIL [5]
> ---|---|---|---|---
> Normalized reward | **24.31($\pm$4.51)** | **23.2($\pm$7.43)** | 0($\pm$0) |  2.47($\pm$0.32)
>
> The result shows that our method is robust to mismatched dynamics.
>
> **References**
>
> [1] D. Yarats et al. Image Augmentation Is All You Need: Regularizing Deep Reinforcement Learning from Pixels. In ICLR, 2021.
>
> [2] M. Laskin et al. Reinforcement Learning with Augmented Data. In NeurIPS, 2020.
>
> [3] Y. J. Ma et al. Smodice: Versatile Offline Imitation Learning via State Occupancy Matching. In ICML, 2022.
>
> [4] Liu. F et al. State Alignment-based Imitation Learning. In ICLR, 2020.
>
> [5] K. Zolna et al. Offline Learning from Demonstrations and Unlabeled Experience. In Offline RL Workshop @ NeurIPS, 2020.

---

> ### Author Response · Authors · 2024-11-25
>
> Dear reviewer hrKU,
>
> Thanks again for your constructive review to improve our paper. As the author-reviewer discussion period is close to its end, we kindly invite you to further consider our response, which we believe has addressed all your concerns raised in the rebuttal. Thank you very much!

---

> > ### Comment · Reviewer_hrKU · 2024-11-26
> > **Official Comment by Reviewer hrKU**
> >
> > Thank you for your response to my questions. I appreciate the experiments you conducted and the modifications you made. Taking into account the contribution and innovation of the paper, I have decided to adjust my score to 5.

---

> > > ### Author Response · Authors · 2024-11-26
> > >
> > > Thanks a lot for appreciating our rebuttal and useful advice for our work!

---

> > > > ### Author Response · Authors · 2024-11-26
> > > >
> > > > Dear reviewer hrkU,
> > > >
> > > > Thanks again for appreciating of our rebuttal. Since there are still a few days before the extended discussion period deadline, is there any remaining concern that we can try to address before the rebuttal period concludes?

---

> ### Author Response · Authors · 2024-12-02
>
> Dear reviewer hrKU,
>
> Thanks again for your effort in providing constructive feedback on improving our paper! As the discussion period will come to an end in less than 24 hours, we would like to know if you have any remaining concern, so that we can address them. We are sincerely looking forward to hearing from you, and are always happy to further discuss with you.

---

### Official Review · Reviewer_ySZG · 2024-11-03

**Soundness:** 2
**Presentation:** 3
**Contribution:** 2
**Rating:** 5
**Confidence:** 3

**Summary:**

This paper proposes a latent Wasserstein adversarial imitation learning (LWAIL) method for achieving expert-level performance with limited state-only expert episodes. The latent space is obtained through a pre-training stage by the Intention Conditioned Value Function (ICVF) model. Experiments on MuJoCo environments demonstrated that LWAIL outperforms prior Wasserstein-based IL methods and prior adversarial IL methods.

**Strengths:**

1.	The proposed LWAIL method contains pre-training and imitation stages. Starting from an illustrative example showing that Euclidean is not a good distance metric, ICVF with random data is used in the pre-training stage to find a more meaningful embedding space. ICVF-trained embedding provides a more dynamics-aware metric than the vanilla Euclidean distance. Then in the imitation stage, the ICVF-learned embeddings are frozen and LWAIL minimizes the 1-Wasserstein distance between the state-embedding-pair occupancy distributions since it allows for a smoother measure and leverages the underlying geometric property of the state space.
2.	The learned reward by LWAIL performs equal to or better than TD3 with ground truth reward.

**Weaknesses:**

1.	This paper claims that the proposed LWAIL can learn more efficient and accurate imitation from limited expert data with only one expert trajectory. The key reasons for this property have not been explained. How do ICVF-trained embeddings contribute to this property?
2.	There are many state embedding methods, however, there is not experimental comparison between ICVF-trained embeddings and other state-of-the-art state embeddings.

**Questions:**

1.	Figure 6 shows the contribution of ICVF-learned embedding. What is the difference between WDAIL and LWAIL without ICVF embedding?
2.	Figure 2 illustrates the same trajectory in the original state space and the embedding space for Hopper and Halfcheetah. It seems ICVF-trained embedding provides a much more dynamics-aware metric than the vanilla Euclidean distance. Is this observation consistent in more environments?

---

> ### Author Response · Authors · 2024-11-22
> **Response to Reviewer ySZG (Part 1 of 2)**
>
> Thanks for your constructive advice. We answer questions next:
>
> **Q1. How do ICVF embeddings contribute to accurate imitation from limited expert data?**
>
> We have discussed this issue in Sec. 3.2, line 203-215. We have modified the paper to highlight this more clearly. To summarize, ICVF embeddings contribute to accurate imitation by creating an embedded state space where the Euclidean distance is more aligned with dynamic differences between states, thus fixing the core problem of prior Wasserstein imitation learning with Kantorovich-Rubinstein (KR) duality: reliance on Euclidean distance between states.
>
> To empirically validate the effectiveness of ICVF embeddings, we follow the challenging setting of prior work (e.g. PWIL) and test our method with limited expert trajectories. Results are shown in the table below. Our method can match the state distribution very well even when the expert data is scarce. Additionally, we have tested our method with multiple expert trajectories, and it has consistently shown excellent performance regardless of the number of expert trajectories.
>
> Env | Hopper | HalfCheetah | Walker | Ant | Average
> ---|---|---|---|---|---
> 1-traj | 110.52 ($\pm$1.06) | 86.71 ($\pm$5.67) | 105.30 ($\pm$2.33) | 80.56 ($\pm$13.09) | 95.77
> 5-traj | 107.65 ($\pm$7.47) | 93.28 ($\pm$1.97) | 107.32 ($\pm$1.36) | 87.23 ($\pm$10.43) | 98.87
> all expert dataset (1M transitions) | 109.34 ($\pm$3.87) | 94.18 ($\pm$3.12) | 104.37 ($\pm$1.97) | 90.81 ($\pm$9.61) | 99.67
>
> **Q2. No comparison between ICVF-embedding and other embeddings.**
>
> While there are embedding methods for RL/IL, many of them are not applicable to our case. For example, most empirical state embedding methods are for visual environments [1, 2] or for cross-domain dynamics matching [3, 4].  Among theoretical state embedding methods, low-rank MDPs are not applicable to the MuJoCo environment, and bi-simulation requires a reward signal which is unavailable in IL.
>
> Nonetheless, we identify two contrastive learning-based baselines most suitable for our scenario: CURL [5] and PW-DICE [6]. Both methods use InfoNCE [7] as their contrastive loss for better state embeddings. Their difference: 1) CURL updates embeddings with an auxiliary loss during online training, while PW-DICE updates embeddings before all other training; 2) CURL compares the current state with different noises added as positive contrast examples, while PW-DICE uses the next states as positive contrast samples. The result is shown below in normalized reward (higher is better):
>
> Env | Hopper | HalfCheetah | Walker | Ant | Average
> ---|---|---|---|---|---
> LWAIL | 110.52 ($\pm$1.06) | 86.71 ($\pm$5.67) | 105.30 ($\pm$2.33) | 80.56 ($\pm$13.09) | **95.77**
> PW-DICE | 110.60 ($\pm$0.77) | 46.07 ($\pm$27.95) | 106.63 ($\pm$1.03) | 85.36  ($\pm$8.12) | 87.16
> CURL | 105.70 ($\pm$1.22) | 87.62 ($\pm$5.10) | 102.97 ($\pm$4.19) | 52.03  ($\pm$8.33) | 87.08
> No Embedding | 108.34 ($\pm$3.42) | 85.98 ($\pm$3.42) | 62.39 ($\pm$20.43) | 40.72 ($\pm$18.95) | 74.36
>
> The result shows that 1) state embeddings generally aid learning; and 2) our proposed method works best.
>
> **Q3. What is the difference between WDAIL and LWAIL without ICVF embedding?**
>
>
> 1. Note, the ICVF embedding is a core part of our LWAIL method, differentiating our work from prior works using Wasserstein imitation learning with KR duality. By adopting an embedding space where the Euclidean distance is aligned with the dynamic differences between the states, we address the core problem of prior works, i.e., the use of an inadequate distance.
>
> 2. This being said, without ICVF embedding, our LWAIL method still differs from WDAIL:
>
> a)  WDAIL needs expert actions, but LWAIL can learn from **action free** expert demonstrations and still outperforms WDAIL as shown in Tab. 1 of the paper. Note, learning from action free expert demonstrations addresses a much harder task due to uncertain environment dynamics [8], and is widely applicable e.g., for learning from video demonstrations (where the expert action is unavailable), or learning from a different embodiment (where the expert action is not applicable).
>
> b) Our LWAIL uses TD3 as the downstream RL algorithm rather than PPO with entropy regularizer as adopted by WDAIL. We ablate the downstream RL algorithm in our LWAIL and show that TD3 outperforms PPO. This is intuitive, as off-policy algorithms are more robust to the change of rewards in the adversarial process.
>
> Env | Hopper | HalfCheetah | Walker | Ant | Average
> ---|---|---|---|---|---
> LWAIL+TD3 (original) | 110.52 ($\pm$1.06) | 86.71 ($\pm$5.67) | 105.30 ($\pm$2.33) | 80.56 ($\pm$13.09) | **95.77**
> LWAIL+PPO | 65.21($\pm$4.81) | 1.02 ($\pm$0.21) | 24.13 ($\pm$2.14) | 9.12 ($\pm$0.85) | 24.87
>
> c) Our method without ICVF embedding differs from WDAIL in technical details. For instance, we propose a normalization in the Wasserstein discriminator’s input $f(\phi(s), \phi(s’)-\phi(s))$ which stabilizes the algorithm.

---

> > ### Author Response · Authors · 2024-11-25
> >
> > Dear reviewer ySZG,
> >
> > Thanks again for your constructive review to improve our paper. As the author-reviewer discussion period is close to its end, we kindly invite you to further consider our response, which we believe has addressed all your concerns raised in the rebuttal. Thank you very much!

---

> ### Author Response · Authors · 2024-11-22
> **Response to Reviewer ySZG (Part 2 of 2)**
>
> **Q4. Is the observation that ICVF-trained embedding provides a much more dynamic-aware metric than Euclidean distance consistent across environments?**
>
> Yes. To verify this, we have updated Fig. 2, to include all MuJoCo environments evaluated in our experiment section and by plotting more steps of a trajectory in each environment. We also plot ground truth reward obtained on each state with different colors (brighter is higher) for a better understanding of the dynamic-aware property. We observe that the high-reward areas are clustered in the embedded latent space, while in the original state space they are scattered. The result shows that we provide a more informative embedding for the agent.
>
> **References**
>
> [1] L. Meng et al. Unsupervised State Representation Learning in Partially Observable Atari Games. In CAIP, 2023.
>
> [2] P. Sermanet et al. Time-Contrastive Networks: Self-Supervised Learning from Video. ArXiv, 2017.
>
> [3] Y. Duan et al. One-Shot Imitation Learning. In NIPS, 2017.
>
> [4] T. Franzmeyer et al. Learn what matters: cross-domain imitation learning with task-relevant embeddings. In NeurIPS, 2022.
>
> [5] A. Srinivas et al. CURL: Contrastive Unsupervised Representations for Reinforcement Learning. In ICML, 2020.
>
> [6] K. Yan et al. Offline Imitation from Observation via Primal Wasserstein State Occupancy Matching. In ICML, 2024.
>
> [7] A. Oord et al. Representation Learning with Contrastive Predictive Coding. ArXiv, 2018.
>
> [8] Z. Zhu et al. Off-Policy Imitation Learning from Observations. In NeurIPS, 2020.

---

> ### Author Response · Authors · 2024-12-02
>
> Dear reviewer ySZG,
>
> Thanks again for your effort in providing constructive feedback on improving our paper! As the discussion period will come to an end in less than 24 hours, we would like to know if you have any remaining concern, so that we can address them. We are sincerely looking forward to hearing from you, and are always happy to further discuss with you.

---

### Author Response · Authors · 2024-11-22
**Summary and Response to Common and Important Questions**

We thank all reviewers, ACs and SACs for their constructive feedback on our work. We are delighted that the reviewers appreciate our paper as well-written with clear objectives (reviewer hrKU), showing a good understanding of the main challenge in imitation learning (reviewer Eovk), and that with simple implementation (reviewer hrKU), our idea allows to leverage geometric properties of the state space (reviewer ySZG) which is inspiring for imitation learning and reinforcement learning (reviewer Eovk).

We have updated our paper based on the feedback, and marked all modified parts using red color. We answer some common and important questions here:

**Q1. How does our metric contribute to better performance?**

The denominator of the 1-Lipschitz constraint forces existing Kantorovich-Rubinstein (KR) duality-based methods to use Euclidean distance in practice due to the gradient regularizer. However, the Euclidean distance is typically not environment dynamics-aware (see illustration in Fig. 1a). Hence, intuitively, we expect an embedding space in which the Euclidean distance is dynamics-aware to yield better performance. We find the ICVF embedding space to satisfy this property because the ICVF training process is inherently dynamics-aware. Indeed, our developed method to incorporate ICVF embeddings shows better performance.

We also provide updated qualitative (Fig. 2) and newly added quantitative (see the next point of our response for details) ablations. Concretely, we also incorporate other embedding spaces in our method, and we study baselines with ICVF embedding. Results show that the embedding can indeed improve the performance but our design performs best when combined with the ICVF embedding.

**Q2. Empirical evaluations.**

We selected vector-based, widely-adopted MuJoCo environments as our main evaluation testbed following many prior works such as WDAIL, IQ-learn and OPOLO. To answer reviewers’ questions we have added many additional ablations, we summarize the additions below:

**1. Different number of expert trajectories.** (reviewer ySzG) The result shows that our method works similarly well regardless of the number of trajectories.

**2. Comparison between ICVF embedding and other contrastive-based embedding methods in our framework.** (reviewer ySzG) The result shows that ICVF embeddings outperform other contrastive-based embeddings.

**3. Ablation on downstream RL algorithm.** (reviewer ySzG) The result shows that TD3 as the downstream RL algorithm outperforms PPO.

**4. Subsampled expert trajectories.** (reviewer hrKU) The result shows that our method works well with highly incomplete expert trajectories. Our method is hence robust.

**5. Existing GAIL-based methods with ICVF embedding.** (reviewer hrKU) The result shows that our method outperforms existing GAIL-based methods with ICVF embedding.

**6. Robustness of LWAIL with mismatched dynamics between expert demonstration and environment.** (reviewer hrKU) The result shows that our method is robust even if dynamics are not well-aligned with the ICVF embedding.

**7. The effect of Sigmoid reward mapping.** (reviewer Eovk) The result shows that naively adding sigmoid reward mapping does not benefit TD3.

Edit: we added another experiment:

**8. Pseudo-reward metric curves.** (reviewer Eovk) The result shows that both sigmoid and ICVF embedding are important for the stability and positive correlation of pseudo-reward generated by $f$ with respect to ground-truth reward.

---

### Meta-Review · Area_Chair_A7PG · 2024-12-20

**Metareview:**

(a) Summary: This paper proposes a latent Wasserstein adversarial imitation learning (LWAIL) method for achieving expert-level performance with limited state-only expert episodes.
(b) Strengths: The paper is generally well-written. The motivation and problem definition is clear
(c) Weaknesses: The reviewers all provided generally negative feedback on the paper. Some major concerns include: the paper lacks novelty, and some technical details are missing.
(d) Some of the reviewers' concerns were not fully addressed. The contributions of the paper are somewhat incremental.

**Additional Comments On Reviewer Discussion:**

Some of the reviewers raised their scores, but all reviewers still felt the paper is below the borderline.

---

### Decision · Program_Chairs · 2025-01-22

Reject